# Variation in the Floral Scent Chemistry of *Nymphaea* ‘Eldorado’, a Valuable Water Lily, with Different Flowering Stages and Flower Parts

**DOI:** 10.3390/plants13070939

**Published:** 2024-03-24

**Authors:** Qi Zhou, Feng Zhao, Man Shi, Huihui Zhang, Zunling Zhu

**Affiliations:** 1College of Environmental Ecology, Jiangsu Open University, Nanjing 210036, China; zhouqi514@njfu.edu.cn; 2College of Landscape Architecture, Nanjing Forestry University, Nanjing 210037, China; zhanghuihui@njfu.edu.cn; 3College of Architectural Engineering, Jiangsu Open University, Nanjing 210036, China; zhaof@jsou.edu.cn; 4State Key Laboratory of Subtropical Silviculture, Zhejiang Agriculture and Forestry University, Hangzhou 311300, China; shiman1031@zafu.edu.cn

**Keywords:** *Nymphae* ‘Eldorado’, volatile organic compound, HS-SPME/GC–MS, floral scent, flowering stage, flower parts

## Abstract

*Nymphaea* ‘Eldorado’, a valuable water lily, is a well-known fragrant plant in China. Studying the temporal and spatial characteristics of the floral components of this plant can provide a reference for the further development and utilization of water lily germplasm resources. In this study, headspace solid-phase microextraction (HS-SPME) combined with gas chromatography–mass spectrometry (GC-MS) was used to explore the types and relative contents of floral components at different flowering stages (S1: bud stage; S2: initial-flowering stage; S3: full-flowering stage; S4: end-flowering stage) and in different floral organs of *N.* ‘Elidorado’, combined with the observation of the microscopic structure of petals. A total of 60 volatile organic compounds (VOCs) were detected at different flowering stages, and there were significant differences in floral VOCs at different flowering stages and in different flower organs. The volatile compounds of *N.* ‘Eldorado’ can be divided into seven chemical classes,, namely, alkenes, alcohols, esters, aldehydes, ketones, alkanes, and others; the most common were alkenes and alkanes. A total of 39, 44, 47, and 42 volatile compounds were detected at S1, S2, S3, and S4. The VOCs present in high concentrations include benzaldehyde, benzyl alcohol, benzyl acetate, trans-α-bergamotene, α-curcumene, cis-α-farnesene, and so on. The types and total contents of volatiles at the full-flowering stage were higher than at other flowering stages. Comparing the VOCs in different parts of flower organs, it was found that the contents of alcohols, esters, and aldehydes were greatest in the petals, the alkenes in stamens were abundant with a relative content of up to 54.93%, and alkanes in the pistil were higher than in other parts. The types and total contents of volatiles in the stamens of *N.* ‘Eldorado’ were higher than those in other flower organs; they were the main part releasing fragrance. The observation of petal microstructure revealed that the size and quantity of the papillae on the epidermises of petals, the number of intracellular plastids, and the aggregates of floral components (osmophilic matrix granules) were significantly higher at the full-flowering stage than at the other flowering stages. This study suggested the main flowering stage and location at which the floral VOCs are released by *N.* ‘Eldorado’ and provided a reference for guiding the breeding of this water lily, exploring genetic patterns and developing related products.

## 1. Introduction

Floral scents are composed of volatile organic compounds (VOCs) with low relative molecular masses, such as terpenoids, phenylpropanoids/benzenoids, and fatty acid derivatives, and include some compounds containing nitrogen or sulfur [1,2]. Floral VOCs have important sensory and aesthetic value in ornamental plants and play a role in attracting insects to ensure pollination and fruiting in all floral plants in general [3]. When plants are invaded by foreign organisms, VOCs can be used to attract their natural enemies and to implement other plant defenses [4,5]. Floral VOC composition, relative VOC content, and VOC release differ among species and varieties, and even within the same variety at different growth stages and environments [6,7].

Research on the floral VOCs of ornamental plants has received extensive attention, mainly involving the identification of plant VOCs, the analysis of the main components of floral scent, and the influence of VOCs on insect pollination behavior [8,9]. In plants, more than 1000 floral volatile compounds have been identified, covering more than 60 families [10]. The common floral scent components found in many plants are linalool, benzyl alcohol, benzaldehyde, ocimene, and benzyl alcohol [11]. The volatile components of the flowers of many ornamental plants, such as *Rosa rugosa* [12], *Lilium* spp. [13], and *Orchidaceae* spp. [14], have been studied in detail. However, there have been few studies on VOCs from water lilies, a well-known aquatic flowering plant. With the development of molecular biology, the anabolic pathway of floral compounds has been gradually explored, as well as several new methods for the genetic improvement of plant floral fragrances [15,16].

There were differences in the types and amounts of VOCs released by different plants during the flowering period. Research has shown that the total volatile content in *Lilium* spp. flowers at the full flowering stage was higher than that at the initial flowering stage, with a high content of linalool, (Z)-β-basil, 1,8-cineole, (+) limonene, laurene, methyl benzoate, etc. [17,18]. The release of VOCs from the flowers of *Jasminum sambac* were generally higher at the full-flowering stage, and α-farnesene, linalool, and benzyl acetate were most abundant in the flowers [19]. *Phalaenopsis bellina* had the highest number of volatiles during the morning and full bloom, with 29.82% monoterpene and 23.33% sesquiterpenes in total [20]. In addition, the content and composition of volatile components in plants are closely related to the flower parts [21]. The relative content of alkanes in the stamens of peony plants is much higher than that in the whole flower and in petals, so stamens may be important sites of alkane synthesis [22]. Other studies have shown that the secretion from flowers is also related to their cell structure. In petals, the fragrance is usually released by diffusion through the entire epidermal cell. For example, the release of fragrance in fragrant plants such as *Styrax tonkinensis* [23] and *Opisthopappus taihangensis* [24] is related to the cell structure of their petals, especially the epidermal cell structure. Glandular trichomes (GTs) are the main organs responsible for the emission of volatiles.

*Nymphaea* L. is a perennial aquatic herb belonging to the *Nymphaea* family, which can be divided into tropical water lilies and hardy water lilies [25]. There are more than 50 species (including varieties) of water lilies worldwide, mainly distributed in tropical, subtropical, and temperate areas [26]. They are widely used in food processing, spice development, garden greening, and other applications [27,28]. The floral scent is one of the most important ornamental traits of water lilies, and its composition and content differ among varieties. At present, compared with the characteristics that are easy to observe, such as flower color, research on *Nymphaea* floral scents has focused mainly on the analysis of the VOCs of different *Nymphaea* spp. and their relationship with putative pollinators [28,29,30], and on the biosynthesis and emission of the major constituent of floral scent [31]; however, little research has been carried out on VOCs at different flowering stages and in the floral organs of water lilies. Zhang et al. [31] found that the flowers of the blue-petal water lily (*N. colorata*) release 11 different volatile compounds, including terpenoids (sesquiterpenes), fatty acid derivatives (methylcaprate), and aromatic compounds. Su et al. [32] studied the volatile components of the flowers of 62 water lily varieties; a total of 72 volatile components were detected with alkenes being most common. Zhou et al. [33] found that alkenes, alcohols, and alkanes were abundant in the flower petals of *N. hybrid*, and the difference in volatile compounds in different *N. hybrids* was closely related to the structure of the perianth.

*N.* ‘Eldorado’ is a tropical water lily with large flowers, golden petals, and pleasant flower fragrances. Chinese horticulturists cross-bred water lilies for many years and cultivated this variety after the introduction of the American fragrant water lily as a breeding material. It not only has high ornamental value but can also be processed into scented tea, which has great potential for development [34,35]. At present, it is cultivated in southern China and is an excellent aromatic plant that is widely used in food, cosmetics, pharmaceuticals, and other industries [36,37]. Although this species of water lily is widely used, the lack of relevant research on the release of floral scent compounds limits its further development and application.

In this study, HS-SPME/GC-MS was used to analyze and identify the VOCs of *N.* ‘Eldorado’ at different flowering stages and in different flower organs. Moreover, the ultrastructures of perianth cells at different flowering stages were observed by scanning electron microscopy (SEM) and transmission electron microscopy (TEM), and the reasons for the floral differences were determined from the perspective of cytology. Through this study, we hope to provide basic data for the development and utilization of aroma substances from water lilies and to lay a foundation for the future breeding and quality identification of *Nympheae* plants.

## 2. Results

### 2.1. Volatile Compounds of the Flower Petals of N. ‘Eldorado’ at Different Development Stages

#### 2.1.1. Chemical Constituents and Contents of Volatile Compounds at Different Flowering Stages of *N.* ‘Eldorado’

The different development stages of *N.* ‘Eldorado’ flower used in this study are shown in Figure 1. A typical GC-MS total ion chromatogram (TIC) for the volatile chemical profile of *N.* ‘Eldorado’ is shown in Appendix A. A total of 60 compounds were detected by the HS-SPME/GC-MS from *N.* ‘Eldorado’ petals at different stages, as shown in Table 1. The types of *N.* ‘Eldorado’ flower components differed among the different stages. A total of 39, 44, 47, and 42 volatile compounds were detected at S1, S2, S3, and S4, respectively, with total relative contents of 96.01%, 97.92%, 98.96%, and 91.64%, respectively. The floral scent consisted mainly of benzaldehyde, benzyl alcohol, benzyl acetate, trans-α-bergamotene, α-curcumene, cis-α-farnesene, pentadecane, α-bisabolene, (Z)-5-dodecenyl-1 acetate, 8-heptadecene, heptadecane, (E6,E9)-heptadecadiene, and heneicosane, which account for more than 72% of the total volatile compound content in *N.* ‘Eldorado’. The types and relative concentrations of VOCs at full-flowering stage were higher than those at the other stages.

A Venn diagram was drawn according to the statistical analysis of the floral volatile composition and content of *N.* ‘Eldorado’ petals at different flowering stages (Figure 2). Twenty-seven compounds were present at all four flowering stages, but their relative contents changed dynamically at different stages. On the other hand, six volatile compounds were uniquely identified at S3, and five were unique at S1 and S4.

The concentration of the major volatile compounds varied at different flowering stages. The content of benzaldehyde, benzyl alcohol, and benzyl acetate was significantly higher at the full-flowering stage than at the other stages. At the bud stage, the aroma scent consisted mainly of trans-α-bergamotene (10.52%), pentadecane (34.13%), and (Z)-5-dodecenyl-1 acetate (8.39%). At the initial flowering stage, the fragrance concentration consisted mainly of benzyl alcohol (8.69%), trans-α-bergamotene (10.68%), pentadecane (22.63%), and (Z)-5-dodecenyl-1 acetate (7.88%). The content of benzaldehyde (5.78%), benzyl alcohol (23.43%), and benzyl acetate (12.15%) was significantly higher at the full-flowering stage than at the other stages (*p* < 0.05), which resulted in a strong fragrance. At the end-flowering stage, the aroma consisted mainly of trans-α-bergamotene (16.19%), pentadecane (21.21%), and (Z)-5-dodecenyl-1 acetate (10.18%), and the aroma was lighter.

#### 2.1.2. Comparison of the Main Volatile Compounds at Different Flowering Stages in *N.* ‘Eldorado’

The VOCs of *N.* ‘Eldorado’ at different flowering stages can be grouped into seven classes of compounds, namely, alkenes, alcohols, esters, aldehydes, ketones, alkanes, and others (Figure 3). Among the seven classes, alkenes (28) accounted for the highest proportion of the total VOCs (47.07%), followed by alkanes (8); the least common were aldehydes and ketones, with only four types each.

There were differences in the contents of seven types of substances at different stages in *N.* ‘Eldorado’. Among them, the relative content of alkenes was the highest at S4 (47.07%), and the relative content of alkanes was the highest at S1 (42.91%). The main alkene was trans-α-bergamotene, which is also a type of sesquiterpene with a woody, warm, tea scent. The relative contents of alcohols, esters, and aldehydes were the highest at S3, and were significantly higher than those at the other stages (*p* < 0.05). At S3, benzyl alcohol (23.43%) was the main alcohol and had a sweet, floral scent; benzyl acetate (12.15%) was the main aldehyde and had a fresh, boiled vegetable scent; and benzaldehyde (5.78%) was the main aldehyde and had a cherry, fruity scent. The total relative contents of alkenes and alkanes were also higher at different flowering stages, accounting for 48%–82% of the total VOCs, indicating that alkenes and alkanes are the two main volatile compounds in *N.* ‘Eldorado’. At the full-flowering stage (S3), VOCs from all classes were present in some proportion, and the distribution was more uniform, so the flowers had a more intense scent at this time.

#### 2.1.3. Hierarchical Cluster Analysis and Principal Component Analysis of the Volatile Compounds

Hierarchical cluster analyses (HCAs) were used to analyze the data for 60 volatile compounds obtained from the four flowering stages of *N.* ‘Eldorado’, and three samples were used to perform analyses for each stage (Figure 4). The 12 *N.* ‘Eldorado’ samples could be divided into four groups. The first group was the S3 group, which was characterized by high concentrations of hexadecane, 2,6,10-trimethyltridecane, methyl salicylate, p-anisaldehyde, 4-methoxybenzyl alcohol, anisyl acetate, benzyl acetate, benzaldehyde, and cis-3-hexenyl hexanoate. The second group included three samples of plants at S1, with high concentrations of tetradecanal, isocaryophyllene, (Z6),(Z9)-pentadecadien-1-ol, 1-methyl-4-(6-methylhept-5-en-2-yl)cyclohexa-1,3-diene, (E)-α-farnesene, pentadecane, and 6-tridecene. The third and fourth groups contained plants at S2 and S4, respectively, and the main volatiles in the S4 group included dihydrocurcumene, ç-terpinene, o-cymene, 1-(1,5-dimethyl-4-hexen-1-yl)-4-methyl-1,3-cyclohexadiene, α-pinene, sabinene, and D-limonene.

Principal component analysis (PCA) was used to identify which volatiles contributed the most to the differences among the different flowering stages of *N.* ‘Eldorado’ using the data of 60 volatile compounds (Figure 5A) and the seven classes of volatile compounds (Figure 5B). Using the factor scores, the tested *N.* ‘Eldorado’ samples were positioned in a two-dimensional space that was divided into four groups based on the scores. As shown in Figure 5A, the volatiles of benzyl alcohol, 1-methoxy-4-methyl-benzene, methyl salicylate, p-anisaldehyde, benzyl acetate, benzaldehyde, 4-methoxybenzyl alcohol, cis-3-hexenyl hexanoate, and anisyl acetate were highly positively correlated with the S3 samples. Isocaryophyllene, (E)-α-farnesene, (6Z,9Z)-pentadecadien-1-ol, 8-heptadecene, pentadecane, heptadecane, 1-methyl-4-(6-methylhept-5-en-2-yl)cyclohexa-1,3-diene, and 2-phenyl-undecane were positively correlated with the S1 samples. However, the S2 group was characterized by only α-ionone. The S4 group was characterized by high amounts of D-limonene, levoverbenone, α-bergamotene, zingiberene, and others. Figure 5B shows that alcohols, esters, and aldehydes were the most important volatile compounds in the S3 samples; alkanes, alkenes, and ketones were the predominant volatiles in the S4, S2, and S1 samples, respectively.

### 2.2. Volatile Components in Different Parts of N. ‘Eldorado’ Flowers

Different parts of the *N.* ‘Eldorado’ flower are shown in Figure 6. The TIC of the volatile chemical profile of *N.* ‘Eldorado’ is shown in Appendix A. A total of 70 compounds were detected in the flowers of *N.* ‘Eldorado’. Forty-seven, fifty-four, and forty-three volatile compounds were detected in petals, stamens and pistils, respectively, with total relative contents of 96.80%, 98.66%, and 92.46%, respectively (Table 2). The contents of alkenes and alkanes are higher in all the different parts. The content of alkenes in stamens was the highest, up to 54.93%, which was significantly higher than that in other parts (*p* < 0.05), and approximately twice that in petals. Stamens may be important synthesis sites for alkene compounds. The contents of alcohols, esters, and aldehydes in the petals were the highest and were significantly higher than those in the other parts (*p* < 0.05). The content of ketones, mainly 2-heptadecanone, in the stamens was high. The types and total relative contents of volatile compounds in the different plant parts were as follows: stamen > petal > pistil.

Figure 7 shows the changes in the eight major volatile components of different parts of the *N.* ‘Eldorado’ flower. The VOCs present in relatively high amounts are trans-α-bergamotene, cis-α-farnesene, 8-heptadecene, (E6,E9)-heptadecadiene, benzyl alcohol, benzyl acetate, benzaldehyde, and pentadecane, which account for more than 70% of the total volatiles in different parts of *N.* ‘Eldorado’. The contents of trans-α-bergamotene and pentadecane in the pistil were the highest and were significantly higher than those in the other parts (*p* < 0.05). The contents of cis-α-farnesene and (E6,E9)-heptadecadiene in stamens were the highest, and were significantly higher than those in other tissues (*p* < 0.05). Benzyl alcohol, benzyl acetate, and benzaldehyde were more abundant in petals than in other tissues, and only 0.41% benzyl alcohol and 0.37% benzaldehyde were detected in the pistil.

### 2.3. Differences in the Cell Structures of the Petals at Different Flowering Stages in N. ‘Eldorado’

The petal structure changes through the stages of *N.* ‘Eldorado’ flower development, and these changes cause the release of VOCs. No papillae were evident in the bud stage of petals of *N.* ‘Eldorado’ (Figure 8A-S1), the cells were closely arranged and were mainly occupied by large vacuoles. The plastids were round or oval, arranged along the cell wall, and a small number of starch granules and osmophilic matrix granules of different shapes and sizes could be seen in the plastids (Figure 8B-S1). At the initial flowering stage, the papillae on the petal surface became more obvious, the number of papillae gradually increased (Figure 8A-S2), and plastids appeared in the cells. The osmophilic matrix granules gradually became round balls with the same shape and size, and gradually spilled out of the cell wall (Figure 8B-S2). At the full-flowering stage, the density and degree of protrusion were higher than those at the other flowering stages (Figure 8A-S3). With increasing cell density, the volume of the plastids gradually increased, but the density of the osmophilic matrix granules inside decreased, and the number of granules outside the cell wall also gradually decreased. Some organelles even disintegrated to varying degrees (Figure 8B-S3). At the end-flowering stage, the papillae on the surface of the petals also gradually shrunk (Figure 8A-S4), the number of internal plastids on the petals significantly decreased, and there were few osmophilic matrix granules inside and outside the cell wall (Figure 8B-S4).

The SEM and TEM results suggest a relationship between cellular structure and the release and loss of the fragrance.

## 3. Discussion

### 3.1. Differences in Floral Components of N. ‘Eldorado’ at Different Flowering Stages

The volatile compounds from flowers are rather complex, and they are affected by different blossom periods [29]. The analysis of VOCs in the flowers of *Nymphaea* L., one of the basic groups of angiosperms, can provide basic data for the study of coevolution between floral and pollinating animals [30]. From the HS-SPME/GC-MS analysis, a total of 60 compounds were detected in the petals of *N.* ‘Eldorado’ at different flowering stages, and 27 compounds were present at all four flowering stages. The types and total relative contents of VOCs decreased in the following order: full-flowering stage > initial-flowering stage > bud stage > end-flowering stage, indicating that the types and relative contents of VOCs at the full-flowering stage were higher. In most plants (such as citrus flowers [38], *Cananga odorata* [39], and *Ocimum citriodorum* [40]), the release of VOCs and its diversity are lower at the bud stage and increase sharply at the blooming stage, then VOCs decline sharply after entering the final flowering stage. For example, more hydrocarbon, esters, and alcohol compounds were detected at the full-flowering stage than at the other flower stages of *C. odorata*; the total number of VOCs at the wilted-flowering stage decreased by nearly half of that at full-flowering stage [40]. These results were similar to those of our study. However, the amount of scent emission and the diversity of floral volatiles were greatest at the initial-flowering stage of *Luculia yunnanensis* [41], which was different from the results of the present study.

Our research of the VOCs of *N.* ‘Eldorado’ at different flowering stages includes alkenes, alcohols, esters, aldehydes, ketones, and alkanes, among which the proportions of alkenes and alkanes are the highest. Tsai et al. [29] found the main volatiles from *N. Caerulea* flower at the early stage of the blossom period were 6,9-heptadecadiene (40.1%), pentadecane(15.5%), 8-heptadecene (15.3%), benzyl acetate (10.4%), and benzyl alcohol (4.4%), which were four common major compounds under different flowering periods. However, in our research, benzyl acetate (12.15%), benzaldehyde (5.78%), and benzyl alcohol (23.43%) reached their highest levels at full-flowering stage; only (E6,E9)-heptadecadiene (6.82%) and cis-α-farnesene (5.11%) were higher at the initial-flowering stage than at the other stages. Other studies have shown that there were no marked differences in floral scent composition between the pistillate (day 1) and staminate phases (day 2) of the flower-blooming period in seven species of *Nymphaea* subg. *Hydrocallis* (Nymphaeaceae), with the exception of *N. rudgeana*, in which the emission of (methoxymethyl)benzene was dramatically reduced during the staminate phase, and floral scent discharge was greater in the pistillate phase than in the staminate phase [30]. Zhou et al. [33] analyzed the chemical compositions of four kinds of *N. hybrid*, and found that the major volatile components were benzyl alcohol, pentadecane, trans-α-bergamotene, (E)-β-farnesene, and (E6,E9)-heptadecadiene, which released the sweeter fruity floral scent. Previous studies found that the main VOCs in tropical water lilies were benzyl alcohol, benzyl acetate, 6(E),8(E)-heptadecadiene, 6,9-heptadecadiene, pentadecane, 8-heptadecene, trans-β-farnesene, and Z,Z-10,12-hexadecadienal, while pentadecane, undecane, cis-ocimene, tridecane, 8-heptadecene, 6,9-heptadecadiene, and tetradecane were the major VOCs in hardy water lilies [32,42]. The main floral scent components at different flowering stages of *N.* ‘Eldorado’ were found to be benzyl acetate, benzyl alcohol, benzaldehyde, trans-α-bergamotene, (E)-β-farnesene, α-bisabolene, and α-sesquiphellandrene. These differences may be related to the experimental materials (water lily varieties), sampling time, testing methods, quality control standards, planting conditions, and environmental climate.

Some studies suggest that, when flowers are in the bud stage, the floral VOCs exist in the form of aroma precursor substances, and when they open, these precursor substances volatilize under the action of enzymes [43,44], which may be one of the reasons why the VOCs are more abundant at the full-flowering stage of *N.* ‘Eldorado’ and reduced at the end-flowering stage, which may be due to the decline of flowers and the decrease of some enzyme activity [45]. Recent studies have shown that plants can attract more pollinators by increasing the volatile emissions of flowers, thus increasing the speed of pollination and mating [46]. The main visiting time of the effective pollinators of *N. hybrid* was concentrated during the blooming stage from 8:00 a.m. to 10:00 a.m., when the flowers are fully blooming with a strong fragrance, which ensures the probability of successful pollination to a certain extent [25]. In this study, the release of VOCs in *N.* ‘Eldorado’ was highest at the full-flowering stage; the plant can attract more pollinators and improve its reproductive efficiency at this time. However, further research is needed on which volatile compounds play a decisive role in insect pollination.

### 3.2. Differences in Petal Structure at Different Flowering Stages and Their Relationships with Floral Release

With the opening and decay of plant flowers, the release of floral scent changes correspondingly, which is closely related to the structure of flower organs [47]. The difference in the volatile compounds of *N. hybrid* was closely related to the structure of the perianth, and the density of protrusion and the number of plastids and osmiophilic matrix granules in the petals play key roles in obtaining the fragrance [33].

In this study, by comparing the ultrastructures of petals at different flowering stages, it was found that the size and number of plastids and osmiophilic matrix granules at the full-flowering stage were significantly higher than those at other stages, which further confirmed that the mastoid protrusions outside the petals and the osmophilic matrix granules inside the petals had a certain synergistic relationship with the formation of the fragrance. A study on the VOCs of *Osmanthus fragrans* flowers also confirmed this view [48]; the aggregates of the floral scent components (osmiophilic matrix granules) of *O. fragrans* formed in the cytoplasts, then overflowed from the cell walls, spilled out through the abundant fold structures in the epidermal cells, and quickly evaporated, leading to speculation that the difference in the ultrastructure of the flower petals may be the reason for the different flower scents of the Osmanthus varieties. Zhang et al. [49] reported that there was a large amount of particulate matter in the upper and lower epidermal cells of the petals of *Jasminum sambac*, which decreased significantly during the opening of jasmine before finally dissolving, suggesting that this particulate matter was related to the formation of the aroma of jasmine. The changes in the petal morphology and structure of *N.* ‘Eldorado’ made the floral scent more abundant and the release quantity higher at the full-flowering stage.

### 3.3. Differences in Volatile Components in Different Parts of N. ‘Eldorado’ Flowers

In most plants, flowers are the main parts releasing aroma compounds; the contributions of various organs and tissues of flowers to the release of aroma substances are not equal, and there are spatial differences [50]. Studies have shown that the fragrance of most plants originates from petals, and other parts, such as the stamen, pistil, calyx, and nectary, can also emit a small amount of fragrance [51,52].

In this study, the types and total relative contents of VOCs in the different parts of the *N.* ‘Eldorado’ flower decreased as follows: stamens > petals > pistils. The stamen contained the most VOCs, which played a dominant role in the whole flower’s volatile compounds. Similar results have also been confirmed for other water lily varieties. Yuan et al. [42] analyzed the floral components of 56 water lily varieties and noted that the stamen and petals of water lily were the main floral organs that released fragrance; the stamen released fragrance, accounting for 70–90% of the whole flower, and was the main part that released VOCs, followed by the petals and pistil. GS-MS analysis of the floral extracts of ‘Paul Blue’ of water lily showed that stamens and petals were the main floral organs involved in fragrance release in this variety [53], and our study further confirmed these conclusions. The content of metabolites in the pistil, stamen, and petal of *Nymphaea* ‘Panama Pacific’ flowers was significantly different, and the levels of vinchnestine and tetrahydroducine in the stamen were relatively high [54]. These studies indicate that the stamen of water lilies can be used as a natural raw material in the field of medicine with certain development potential.

Previous studies have shown that the expression levels of plant floral substance biosynthesis genes are usually consistent with the main parts of floral substance production [55]. Mao et al. [56] investigated the floral VOCs metabolism pathways and differentially expressed genes (DEGs) involved in the biosynthesis of terpenoid compounds in the flower organs of tropical water lily *N.* ‘Paul Stetson’, and revealed that the gene expression patterns of stamens and petals were significantly different from those of pistils. The expression levels of HMGR and DXS, the key genes involved in the synthesis of farnesal and and diterpene kaurene, were higher in both petals and stamens than in pistils. It was speculated that the differences in the gene expression patterns of the three flower organs might be important factors leading to the significant differences in the contents of the aroma substances released.

In the future, it will be possible to further study the changes in the ultrastructure and cell contents of stamens and pistils and explore the key regulatory genes related to the synthesis of flowers via molecular biological methods to further study the relationship between flowers and the structure of flower organs and provide scientific references for the application and research of *N.* ‘Eldorado’ flowers.

## 4. Materials and Methods

### 4.1. Plant Materials

Fresh flowers were obtained at the four different flowering stages (S1: bud stage; S2: initial flowering stage; S3: full-flowering stage; S4: end-flowering stage) of *N.* ‘Eldorado’ (Figure 1). The *N.* ‘Eldorado’ flowers used in the experiments originated from the Germplasm Resource Nursery of Water Lily in Yancuo Village (24°47′ N, 116°41′ E), Zhangzhou, Fujiang Province, China. This area is located in a subtropical humid monsoon climate zone. The average annual precipitation is approximately 1557 mm, the average annual temperature is approximately 22 °C, and rainfall and sunshine are abundant, which are very suitable conditions for the growth of tropical water lilies. A voucher specimen (No.20181025) was deposited in the Institute of Nanjing Forestry University. The *N.* ‘Eldorado’ plants were planted in the same pond under the same cultivation conditions (including irrigation, soil, fertilization, and pest control measures). Three fresh flowers representing three samples from different flowering stages of *N.* ‘Eldorado’ were collected in the morning (approximately 8:00 am) and immediately inserted into deionized water before being transported to the Advanced Analysis Testing Center (AATC) at Nanjing Forestry University for further experiments.

### 4.2. Microstructure of Petals with SEM and TEM

Scanning electron microscopy (SEM): Petal samples collected from *N*.’ Eldorado’ plants at different flowering stages (*n* = 3 × 4 individuals, total = 12) were put into Karnovsky fixative (4% p-formaldehyde, 5% glutaraldehyde in 0.1 M phosphate buffer, pH 7.2), vacuumed under reduced pressure, allowed to thaw, and fixed in a refrigerator at 4 °C for 48 h. Then, a gradient series of ethanol dehydration, vacuum drying, bonding, and coating with gold were performed using an ion sputtering apparatus (E1010, Hitachi, Tokyo, Japan). Finally, the sections were observed and photographed under a QUANTA200 environmental scanning electron microscope (Quanta 200, FEI, Hillsborough, OR, USA) [57]. All the chemicals were purchased from Sinopharm Chemical Reagent Co., Ltd., Beijing, China. Three replicates were conducted for each experiment.

Transmission electron microscopy (TEM): First, petal samples from the full-flowering stage of *N.* ‘Eldorado’ were treated as rectangular pieces with a length of 5 mm and a width of 1 mm and were then fixed in 4% glutaraldehyde prepared with 0.2 mol/L phosphoric acid buffer at pH = 7.2 at 4 °C for 48 h. The sample was rinsed with phosphate buffer 3 times, left for 10~20 min each time, then 3 h postfixation was performed in 1% (*w*/*w*) osmic acid in the same phosphate buffer, which was followed by dehydration with a graded series of acetone and embedding in Epon 812 epoxy resin. Finally, the embedded blocks were sliced with an LKB III ultramicrotome (LKB Instruments, Inc., AB Bromma, Stockholm, Sweden). The slices were stained with uranyl acetate and lead citrate, then photographed under a transmission electron microscope (Hitachi HU 12 A; Hitachi High-Technologies, Tokyo, Japan) [23].

### 4.3. Analysis of Volatile Components

Floral VOCs were extracted from the headspace of 2.0 g of fresh flower samples (petals/stamen/pistil) placed into a 40 mL glass screw top vial (specifications) for 50 min at 45 °C using a 65 μm PDMS/DVB SPME fiber (Supelco Inc., Bellefonte, PA, USA) with manual handle. After extraction, the fiber was desorbed for 3 min at 250 °C in splitless mode in the injection port, and gas chromatography-mass spectrometry was performed by Trace 1300 ISQ LT (Thermo Scientific, Waltham, MA, USA). The SPME fiber was conditioned at 250 °C for 30 min before first use. A DB-5 MS capillary column (5% phenylmethylsiloxane, 30 m × 0.25 mm × 0.25 μm; Thermo Fisher Scientific, Sunnyvale, CA, USA) was used to conduct the analyses. Helium was used as a carrier gas with a flow rate of 1.0 mL/min. The injector temperature was 250 °C in splitless mode. The oven temperature was programmed at 40 °C for 2 min, then raised at 4 °C/min to 110 °C and held for 2 min. Then the temperature was raised at 3 °C/min to 150 °C and held for 2 min, then increased at 5 °C/min to 200 °C and held for 4 min. The flow rate was 1.0 mL/min. The mass selective detector was used in EI mode at an ionization voltage of 70 eV, and the scanning range was *m*/*z* 33–450 in the full-scan mode. The ion source and quadrupole temperatures were 250 °C.

Retention indices (RIs) were calculated by using the retention times of C7–C30 n-alkanes (BNCC, Xinyang, China) according to previously reported methods [33,58]. The volatile components were identified by comparison of their RI and mass spectra with the NIST (National Institute of Standards and Technology) [59], PubChem [60], and Pherobase [61] databases [62,63], and the relevant literature [29,30,42,64,65]. The relative content of each component in the sample was calculated using the peak area normalization method [66]. The spectrum of each compound was analyzed by Xcalibur and NIST 2014 (NIST Database, ChemSW, Inc., Fairfield, CA, USA), and compounds were identified according to their retention times (RTs) and the NIST database.

### 4.4. Statistical Analysis

All the data were analyzed using the SPSS statistical package version 22.0 (IBM Corp., Armonk, NY, USA). Duncan’s multiple range test was used to determine the significant differences between samples (*p* < 0.05). Principal component analysis (PCA) was carried out using Canoco 4.5 software (Microcomputer Power, Ithaca, NY, USA). Hierarchical clustering analysis (HCA) was performed with TBtools software 2022 (https://github.com/CJ-Chen/TBtools/releases (accessed on 23 September 2023)), and the heat map was normalized using the row scale method.

## 5. Conclusions

To our knowledge, this is the first study reporting the presence of floral volatile components in combination with the observation of the microscopic structure of petals at different flowering stages and the detection of VOCs in the different floral organs of *N.* ‘Eldorado’, a valuable water lily. We found that the volatile component emissions from flowers varied among the different flowering stages and different floral organs of *N.* ‘Eldorado’. A total of 60 volatile components at different flowering stages and 70 VOCs were detected in the different floral organs of these plants and were identified and quantified via HS-SPME/GC–MS. The morphology and quantity of the epidermal prominens, intracellular plastids, and matrix particles of *N.* ‘Eldorado’ were closely related to the formation of the floral fragrance. The floral scent components of this water lily are dominated by high levels of alkenes, alcohols, and alkanes. Moreover, further studies are needed to understand the chemical origin of the morphological characteristics to reveal the dissimilarities of the primary and secondary metabolites of these plants. Our results can lead to the investigation of the floral scents of *N.* ‘Eldorado’, which are highly important for the future development and utilization of this water lily as an excellent aromatic plant for the cosmetic, food, and landscaping industries.

## Figures and Tables

**Figure 1 plants-13-00939-f001:**
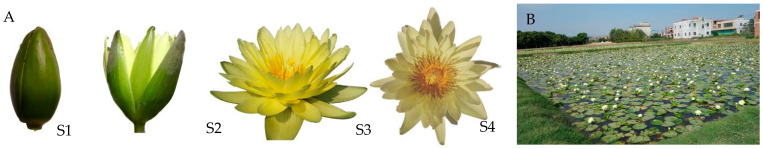
Different development stages of *N.* ‘Eldorado’ flower used in this study. (**A**): (S1) Bud stage; (S2) initial-flowering stage; (S3) full-flowering stage; and (S4) end-flowering stage. (**B**): Living habit of *N.* ‘Eldorado’.

**Figure 2 plants-13-00939-f002:**
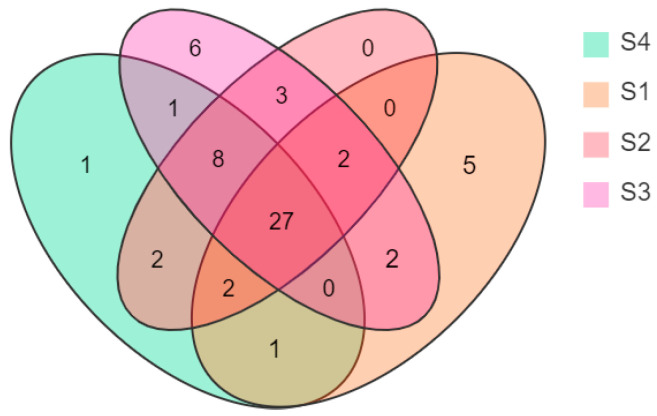
Venn diagram showing the numbers of volatiles at different flowering stages of *N.* ‘Eldorado’ flowers.

**Figure 3 plants-13-00939-f003:**
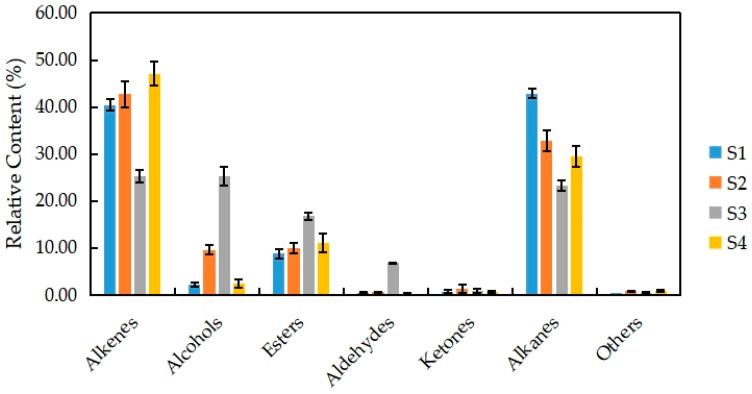
Percentages of alkenes, alcohols, esters, aldehydes, ketones, alkanes, and other volatile compounds released from different development stages of *N.* ‘Eldorado’ flower.

**Figure 4 plants-13-00939-f004:**
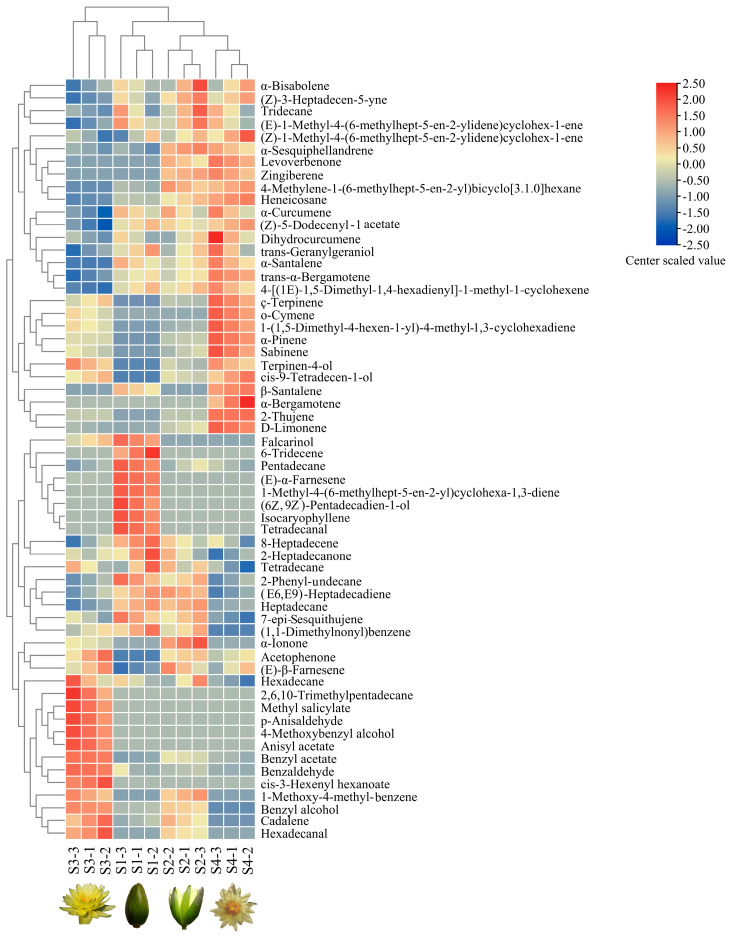
Hierarchical clustering analysis (HCA) and heatmap of volatile compound levels at four flowering stages of *N.* ‘Eldorado’. Values of all studied volatile compounds per stage are shown on the heatmap, colors of the heatmap cells indicate a low (blue) and high (red) abundance of a particular compound. The HCA and dendrogram of samples were according to Euclidean distances. S1-1, S1-2, and S1-3 indicate the three samples of S1, and the same is true of S2, S3 and S4; the same labels are used in Figure 5.

**Figure 5 plants-13-00939-f005:**
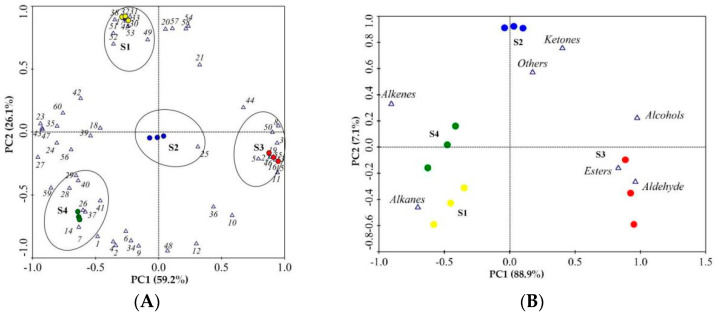
Principal component analysis (PCA) of 12 samples based on PC1 and PC2 scores. The VOCs numbered 1–60 are represented by 1–60 in Table 1. (**A**) Loading plot of samples and the 60 volatile compounds; the ellipses represented samples of the same stage and the VOCs that were highly correlated with that stage. (**B**) Loading plot of samples and the seven different classes of volatile compounds.

**Figure 6 plants-13-00939-f006:**
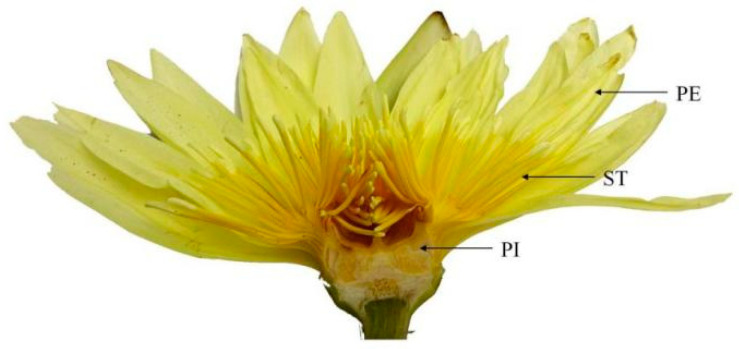
Different parts of *N.* ‘Eldorado’ flower. PE—petal; ST—stamen; PI—pistil.

**Figure 7 plants-13-00939-f007:**
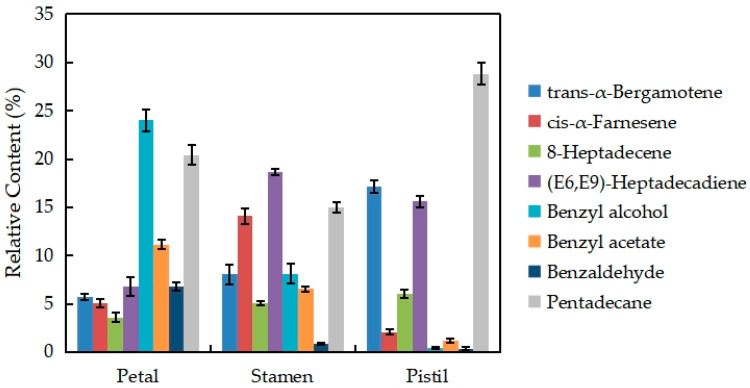
Changes in the eight major volatile components of different parts of *N.* ‘Eldorado’ flowers.

**Figure 8 plants-13-00939-f008:**
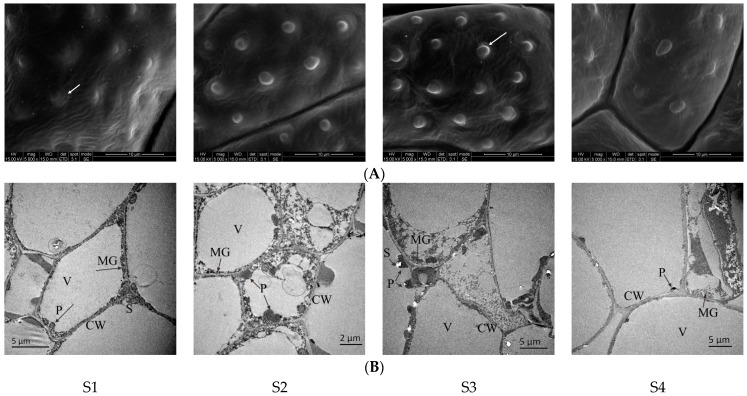
Scanning electron microscopy (SEM, **A**) images and transmission electron microscopy (TEM, **B**) images of *N.* ‘Eldorado’ flower petals at different flowering stages. Arrows in (**A**) represent papillae on the epidermis of flower petals. V—vacuole, CW—cell wall, P—plastids, S—starch granules, MG—Osmophilic matrix granules.

**Table 1 plants-13-00939-t001:** Volatile compounds identified at different stages of *N*. ‘Eldorado’ flower development.

NO.	RI (exp.)	RI (lit.)	Compounds	Molecular Formula	Aroma Description ^1^	Relative Content (%) ^2^			
						S1	S2	S3	S4
1	938	930	2-Thujene	C_10_H_16_	Woody, green, herbal	-	0.07 ^b^	0.08 ^b^	0.38 ^a^
2	940	946	α-Pinene	C_10_H_16_	Pine, turpentine	-	0.17 ^c^	0.23 ^b^	0.71 ^a^
3	958	963	Benzaldehyde	C_7_H_6_O	Cherry, fruity	0.14 ^c^	0.38 ^b^	5.78 ^a^	0.08
4	975	977	Sabinene	C_10_H_16_	Pepper, turpentine, woody	-	0.19 ^c^	0.25 ^b^	0.87 ^a^
5	1002	1005	1-Methoxy-4-methyl-benzene	C_8_H_10_O	Nutty	-	0.14 ^a^	0.15 ^a^	-
6	1030	1022	o-Cymene	C_10_H_14_	Solvent, gasoline, citrus	-	-	0.12 ^b^	0.31 ^a^
7	1032	1198	D-Limonene	C_10_H_16_	Citrus, mint	0.20 ^d^	0.60 ^b^	0.28 ^c^	2.44 ^a^
8	1035	1037	Benzyl alcohol	C_7_H_8_O	Sweet, flower	1.80 ^c^	8.69 ^b^	23.43 ^a^	0.12 ^d^
9	1052	1058	ç-Terpinene	C_10_H_16_	Lemon	-	0.21 ^c^	0.45 ^b^	0.97 ^a^
10	1065	1067	Acetophenone	C_8_H_8_O	Must, flower, almond	-	0.30 ^b^	0.39 ^a^	0.20 ^c^
11	1170	1166	Benzyl acetate	C_9_H_10_O_2_	Fresh, boiled vegetable	0.42 ^d^	2.12 ^b^	12.15 ^a^	0.86 ^c^
12	1162	1164	Terpinen-4-ol	C_10_H_18_O	Turpentine, nutmeg	-	0.10 ^b^	0.26 ^a^	0.25 ^a^
13	1188	1193	Methyl salicylate	C_8_H_8_O3	Peppermint	-	-	0.09 ^a^	-
14	1215	1220	Levoverbenone	C_10_H_14_O	Camphor, menthol	-	0.09 ^b^	-	0.13 ^a^
15	1245	1252	p-Anisaldehyde	C_8_H_8_O_2_	Mint, sweet	-	-	0.60 ^a^	-
16	1252	1247	4-Methoxybenzyl alcohol	C_8_H_10_O_2_	Sweet, powdery creamy	-	-	0.11 ^a^	-
17	1285	1282	6-Tridecene	C_13_H_26_	-	0.06 ^a^	-	-	-
18	1300	1300	Tridecane	C_13_H_28_	Alkane	0.45 ^b^	0.55 ^a^	0.32 ^c^	0.48 ^b^
19	1382	1370	cis-3-Hexenyl hexanoate	C_12_H_22_O_2_	Fruit, prune	-	-	0.10 ^a^	-
20	1390	1393	7-epi-Sesquithujene	C_15_H_24_	-	1.48 ^a^	1.26 ^b^	0.83 ^c^	0.53 ^d^
21	1402	1400	Tetradecane	C_14_H_30_	Alkane	0.48 ^a^	0.39 ^b^	0.44 ^a^	0.31 ^b^
22	1415	1413	Anisyl acetate	C_10_H_12_O_3_	Floral, anisic, fruity	-	-	0.87 ^a^	-
23	1418	1420	α-Santalene	C_15_H_24_	-	0.23 ^a^	0.19 ^b^	-	0.28 ^a^
24	1420	1422	β-Santalene	C_15_H_24_	Woody	0.40 ^b^	-	-	0.70 ^a^
25	1425	1426	α-Ionone	C_13_H_20_O	Woody, violet	-	0.39 ^a^	0.12 ^b^	-
26	1435	1438	α-Bergamotene	C_15_H_24_	Woody, warm, tea	-	-	-	0.17 ^a^
27	1440	1442	trans-α-Bergamotene	C_15_H_24_	Woody, warm, tea	10.52 ^b^	10.68 ^b^	5.74 ^c^	16.19 ^a^
28	1442	1445	4-Methylene-1-(6-methylhept-5-en-2-yl)bicyclo[3.1.0]hexane	C_15_H_24_	-	0.32 ^b^	1.68 ^a^	-	1.75 ^a^
29	1445	1448	Dihydrocurcumene	C_15_H_24_	-	0.27 ^b^	0.28 ^b^	0.21 ^c^	0.36 ^a^
30	1450	1455	Isocaryophyllene	C_15_H_24_	Woody, spicy	0.06 ^a^	-	-	-
31	1458	1460	(E)-α-Farnesene	C_15_H_24_	Woody, sweet	3.29 ^a^	-	0.07 ^b^	-
32	1465	1468	(6Z,9Z)-Pentadecadien-1-ol	C_15_H_28_O	-	0.09 ^a^	-	-	-
33	1468	1472	1-Methyl-4-(6-methylhept-5-en-2-yl)cyclohexa-1,3-diene	C_15_H_24_	-	0.93 ^a^	-	-	-
34	1470	1475	1-(1,5-Dimethyl-4-hexen-1-yl)-4-methyl-1,3-cyclohexadiene	C_15_H_24_	-	-	0.12 ^c^	0.42 ^b^	1.06 ^a^
35	1480	1482	α-Curcumene	C_15_H_24_	-	3.09 ^b^	3.02 ^b^	1.77 ^c^	3.32 ^a^
36	1488	1485	(*E*)-β-Farnesene	C_15_H_24_	Citrus, green	2.76 ^c^	5.11 ^a^	5.07 ^a^	4.17 ^b^
37	1498	1495	Zingiberene	C_15_H_24_	Spicy, fresh, sharp	-	1.22 ^b^	-	1.39 ^a^
38	1503	1500	Pentadecane	C_15_H_32_	Alkane	34.13 ^a^	22.63 ^b^	20.40 ^c^	21.21 ^c^
39	1510	1505	α-Bisabolene	C_15_H_24_	Balsamic	2.27 ^b^	2.73 ^a^	1.83 ^c^	2.46 ^a^
40	1516	1518	(Z)-1-Methyl-4-(6-methylhept-5-en-2-ylidene)cyclohex-1-ene	C_15_H_24_	-	0.13 ^c^	0.16 ^b^	0.11 ^c^	0.20 ^a^
41	1520	1523	α-Sesquiphellandrene	C_15_H_24_	Sweet, fruit, herbal	0.73 ^c^	2.82 ^a^	0.79 ^c^	2.21 ^b^
42	1528	1531	(E)-1-Methyl-4-(6-methylhept-5-en-2-ylidene)cyclohex-1-ene	C_15_H_24_	-	0.21 ^a^	0.23 ^a^	0.13 ^c^	0.19 ^b^
43	1536	1539	4-[(1E)-1,5-Dimethyl-1,4-hexadienyl]-1-methyl-1-cyclohexene	C_15_H_24_	-	0.18 ^a^	0.18 ^a^	0.06 ^b^	0.21
44	1602	1600	Hexadecane	C_16_H_34_	Alkane	0.18 ^b^	0.19 ^b^	0.21 ^a^	0.15 ^c^
45	1612	1615	Tetradecanal	C_14_H_28_O	-	0.27 ^a^	-	-	-
46	1635	1630	2,6,10-Trimethylpentadecane	C_18_H_38_	-	-	-	0.06 ^a^	-
47	1658	1656	(Z)-5-Dodecenyl-1 acetate	C_14_H_26_O_2_	-	8.39 ^b^	7.88 ^c^	3.62 ^d^	10.18 ^a^
48	1665	1667	cis-9-Tetradecen-1-ol	C_14_H_28_O	-	-	0.70 ^c^	1.27 ^b^	1.91 ^a^
49	1675	1667	(E6,E9)-Heptadecadiene	C_17_H_32_	-	6.44 ^a^	6.82 ^a^	3.14 ^b^	2.54 ^c^
50	1680	1672	Cadalene	C_15_H_18_	-	0.05 ^c^	0.15 ^b^	0.23 ^a^	-
51	1692	1680	8-Heptadecene	C_17_H_34_	Alkane	6.85 ^a^	4.85 ^b^	3.59 ^c^	3.97 ^c^
52	1703	1700	Heptadecane	C_17_H_36_	Alkane	5.77 ^a^	5.79 ^a^	1.41 ^c^	1.83 ^b^
53	1712	1715	2-Phenyl-undecane	C_17_H_28_	-	0.74 ^a^	0.57 ^b^	0.30 ^c^	0.28 ^c^
54	1740	1745	(1,1-Dimethylnonyl)benzene	C_17_H_28_	-	0.11 ^a^	0.08 ^b^	0.06 ^b^	-
55	1785	1797	Hexadecanal	C_16_H_32_O	Cardboard	-	0.20 ^b^	0.40 ^a^	-
56	1838	1840	(Z)-3-Heptadecen-5-yne	C_17_H_30_	-	0.36 ^c^	0.57 ^a^	0.18 ^d^	0.49 ^b^
57	1880	1883	2-Heptadecanone	C_17_H_34_O	-	0.74 ^a^	0.52 ^b^	0.43	0.34 ^d^
58	1992	1997	Falcarinol	C_17_H_24_O	-	0.13 ^a^	-	0.07 ^b^	-
59	2102	2100	Heneicosane	C_21_H_44_	Alkane	1.16 ^c^	2.74 ^b^	0.25 ^d^	5.25 ^a^
60	2195	2201	trans-Geranylgeraniol	C_20_H_34_O	-	0.18 ^a^	0.16 ^b^	0.09 ^c^	0.19 ^a^
Total						96.01 ^c^	97.92 ^b^	98.96 ^a^	91.64 ^d^

RI (exp.): Experimental retention indices; RI (lit.): literature retention indices (PubChem, NIST, and the Pherobase). ^1^ Aroma descriptions were obtained from the literature (http://www.thegoodscentscompany.com/ (accessed on 23 September 2023); http://www.scent.net/ (accessed on 23 September 2023); and Pherobase); -: not detectable. ^2^ Statistical analyses were performed with Duncan’s multiple range test. Means with different letters (a, b, c, d) within a row are significantly different at *p* < 0.05. VOCs with very low relative content (<0.05%) and the unidentified compound were not counted in the table.

**Table 2 plants-13-00939-t002:** Classification of the volatile components in different parts of *N.* ‘Eldorado’ flowers.

Categories	Different Parts of the Flower					
	Petal		Stamen		Pistil	
	Number	Relative Content (%)	Number	Relative Content (%)	Number	Relative Content (%)
Alkenes	20	27.86 ^b^	24	54.93 ^a^	21	52.46 ^a^
Alcohols	5	24.54 ^a^	4	8.63 ^b^	4	2.49 ^c^
Esters	5	12.29 ^a^	5	7.24 ^b^	3	1.36 ^c^
Aldehydes	3	7.40 ^a^	5	2.00 ^b^	3	0.64 ^c^
Ketones	3	0.94 ^b^	3	2.85 ^a^	2	0.82 ^b^
Alkanes	8	23.36 ^b^	9	21.84 ^b^	8	34.09 ^a^
Others	3	0.41 ^b^	4	1.17 ^a^	2	0.60 ^b^
Total	47	96.80 ^b^	54	98.66 ^a^	43	92.46 ^c^

Means with different letters (a, b, c) within a row are significantly different at *p* < 0.05.

## Data Availability

Data are contained within the article.

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
