# Peer review of "Variation in the Floral Scent Chemistry of Nymphaea ‘Eldorado’, a Valuable Water Lily, with Different Flowering Stages and Flower Parts"

_plants, 2024, doi:10.3390/plants13070939_

Round 1

Reviewer 1 Report

Comments and Suggestions for Authors

The authors present a well-designed and conducted study on the floral scent or VOCs of a species (or variety?) of water lily. There is surprisingly little know about the genus in general and the research is an important contribution to the field. It clearly meets the remit of plants and is very much publishable.

The manuscript, however, does not justice to the work and needs extensive revision to be publishable.

Overall the style is overly verbose, lacks focus and contains unclear and imprecise statements and formulation.

More specifically:

Introduction:

Generally, the introduction provides a reasonable rational but suffers from imprecise formulation and sweeping statements. Please revise the section to make it a) more focused, b) more precise and quantitative and formulate clear aims and objectives that can be addressed in Results and Discussion.

Specifically:

Avoid using the term ‘aromatic substances’ this is easily confused with aromatic compounds, which comprise a distinct class of chemical compounds.

Equally, be consistent in the terminology, floral VOCs are called, aromatic substances, floral substances, floral fragrance etc.

Make clearer what substances are talked about e.g. lines 64-66:

“Research on the floral substances of ornamental plants has received extensive attention, mainly involving the identification of plant volatiles, the analysis of main floral components, and the influence of VOCs on insect pollination behavior [8,9].”

Does ‘floral substances’ include also non-volatile compounds? How relevant is this in the context?

Avoid sweeping statements that do not contribute any concrete fact or knowledge, e.g. lines 76-78

“The release of volatile components from flowers occurs differently throughout the opening of the flowers, and the amount of volatile compounds released also varies to a certain extent.”

Better phrase this in quantitative terms, starting with the subsequent examples straight away.

Similarly line 79-80: “…Phalaenopsis bellina [19] showed different changes at different flowering stages,…”

What exactly was different and what were the stages?

Equally lines 93-95:

“Due to the important role of floral fragrances in the self-protection and ornamental quality improvement in plants, the study of floral fragrances has received increasing attention and has become a hot topic in basic plant research.”

What is meant by ‘self-protection’? The meaning of the term itself and its important role is not clear from the preceding text.

Lines 106-110:

“At present, compared with the characteristics that are easy to observe, such as flower color and flower type, research on Nymphaea fragrances has focused mainly on the physiological and biochemical content of the extraction process optimization of aroma volatiles [29], aroma component analysis [28], and in vitro activity evaluation of volatile oil extracts [30];”

This sentence is very confusing – what is the physiological content of a fragrance? What is the relevance of process optimisation? Isn’t ‘aroma component analysis’ not simply an analysis of VOCs? What are ‘volatile oil extracts’? There are essential oils, which are extracts and components of essential oils, which are (mostly) VOCs. Avoid confusing the reader by using multiple and unusual terms for the same thing (as mentioned earlier) and instead of excepted terminology resp.

It would also help in this section to clarify the phylogeny of the species used and varieties/cultivars used, e.g.  Nymphaea eldorado  (by the way there is a persistent typo in the manuscript ‘eidorado’ instead of ‘eldorado’)

Results:

Overall, the section lacks focus – what is the main ‘take-home’ message? For example, the Venn diagram already shows components that are unique for each flowering stage – what is the purpose of the cluster analysis or PCA? What is the additional value/information that can be gained?

Figures 2 and 9 do not provide any information beyond that presented in tables and other figures. I would leave those out, the more as the relative abundance scale makes absolute differences invisible.

Keep information of VOCs together – the SEM and TEM data are better presented after it is clear which part of the flower releases more VOCs

Line 158 – from the methods it appears that peak areas were normalised and should add up to 100 % for all four stages.

Line 159-160: It cannot be concluded that the content is larger in S3 when using normalised data. Equally between sample comparisons are valid but only show that relative concentrations are larger, which could be because e.g. fewer compounds are detected, other components are less abundant or the compound is actually more or less abundant. The total peak area provides an indication of overall increase or decrease of abundance of VOC.

Line 174: unclear, what are “differences in the amount of substances with high volatile components”? Substances are all VOCs and the components mentioned in the following do not have particularly high vapour pressures.

Line 193 and following: this section is better placed earlier as it provides an overview of chemical classes and their relative abundance.

Figure 1 – images of stages are in wrong order – S2 is last

Discussion:

This section needs to be more concise and precise – avoid or be careful relating the work to areas that are not directly supported by your research, e.g. phylogeny, genetics, rapid changes in VOCs.

Compare specific details, e.g. number of compounds, kind of compounds and avoid long paragraphs without references.

Lines: 315-320: true but the current research only looked at one species/cultivar.

Line 339: it is not clear why being a large tropical water lily is related to its VOC content and characteristics.

Line 339-362: this is essentially a repetition of results as there is no reference to the literature in the entire section.

Line 363: as earlier be consistent in terminology use ‘floral VOCs’ of VOCs – what constitutes the ‘aroma’?

Material and Methods

The section is verbose in places – be more concise and precise

Line 485 and 495: delete ‘observations’

Line 498: what is meant by ‘The sample was lowered and vacuumed to sink,..”?

Line 499: The sample was rinsed with phosphate buffer not the buffer

Line 500: what acid was added? Was it added to make up to 0.1 % or was the concentration of added acid 0.1 %? What was the percentage of – volume or weight?

Line 509: the fibre was conditioned not aged

In terms of being more concise for example lines 508 to 517 could be written as

“Floral VOC extraction: Floral VOCs were extracted from the headspace of 2.0 g of fresh flower petals placed into a 40 mL glass screw top vial (specifications) for 50 min at 45°C using a 65 µm PDMS/DVB SPME fibre (Supelco Inc, Bellefonte, USA) with manual handle.

Samples were desorbed for 3 min at 2XX °C (specify!) in splitless mode (I assume?) in the injection port of a GC-MS for analysis (Trace 1300 ISO-LT GC1300, Thermofisher Scientific, Sunnyvale, CA, USA).

Fibres were conditioned at 250 °C for 30 min prior to use.”

How were VOCs collected from stamen and pistil? Were stamen and pistils separated from flowers (like the petals that are described)? Was the same weight used, same equipment etc?

Comments on the Quality of English Language

The English is overall of good quality but very verbose and inconsistent in technical terminology and jargon. It lacks conciseness, precision and consistence.

Author Response

Dear Reviewer:

Thank you for your comments concerning our manuscript entitled “Variation in the Floral Scent Chemistry of Nymphaea ‘Eldorado’, a Valuable Water Lily, With Different Flowering Stages and Flower Parts” (plants-2885832). Those comments are all valuable and very helpful for revising and improving our paper, as well as the important guiding significance to our researches. We have studied comments carefully and have made correction which we hope meet with approval. Revised portion are highlighted in the paper.

The main corrections in the paper and the responds to the reviewer’s comments are as following:

Response to Reviewer 1 Comments

Overall the style is overly verbose, lacks focus and contains unclear and imprecise statements and formulation.

More specifically:

Introduction:

Generally, the introduction provides a reasonable rational but suffers from imprecise formulation and sweeping statements. Please revise the section to make it a) more focused, b) more precise and quantitative and formulate clear aims and objectives that can be addressed in Results and Discussion.

Specifically:

Comments 1: Avoid using the term ‘aromatic substances’ this is easily confused with aromatic compounds, which comprise a distinct class of chemical compounds.

Equally, be consistent in the terminology, floral VOCs are called, aromatic substances, floral substances, floral fragrance etc.

Response: We have removed the term ‘aromatic substances’, and were consistent in the terminology as suggested.

Comments 2: Make clearer what substances are talked about e.g. lines 64-66:

“Research on the floral substances of ornamental plants has received extensive attention, mainly involving the identification of plant volatiles, the analysis of main floral components, and the influence of VOCs on insect pollination behavior [8,9].”

Does ‘floral substances’ include also non-volatile compounds? How relevant is this in the context?

Response: The ‘floral substances’ in this paper mainly refer to the floral VOCs, and didn’t include the non-volatile compounds, and we have made corresponding modifications in the article.

Comments 3: Avoid sweeping statements that do not contribute any concrete fact or knowledge, e.g. lines 76-78

“The release of volatile components from flowers occurs differently throughout the opening of the flowers, and the amount of volatile compounds released also varies to a certain extent.”

Better phrase this in quantitative terms, starting with the subsequent examples straight away.

Similarly line 79-80: “…Phalaenopsis bellina [19] showed different changes at different flowering stages,…”

What exactly was different and what were the stages?

Response: We have revised the statements as suggested, and described the main differences. This sentence has been rewritten as “There were differences in the types and amounts of VOCs released by different plants during flowering period. Research has shown that the total volatile content in Lilium spp. flowers at the full flowering stage was greater than that at the initial flowering stage. The components with higher volatile contents were linalsol, (Z)-β-basil, 1,8-cineole, (+) limonene, laurene, methyl benzoate, etc. [17,18]. The release of VOCs from the flowers of Jasminum sambac were generally greater at the full-flowering stage, and α-farnesene, linalool, and benzyl acetate were most abundant in the flowers [19]. Phalaenopsis bellina had the highest number of volatiles during the morning and full bloom, with 29.82% monoterpene and 23.33% sesquiterpenes accounted in total [20]. ”

Comments 4: Equally lines 93-95:

“Due to the important role of floral fragrances in the self-protection and ornamental quality improvement in plants, the study of floral fragrances has received increasing attention and has become a hot topic in basic plant research.”

What is meant by ‘self-protection’? The meaning of the term itself and its important role is not clear from the preceding text.

Response: We have deleted these sweeping and irrelevant statements.

Comments 5: Lines 106-110:

“At present, compared with the characteristics that are easy to observe, such as flower color and flower type, research on Nymphaea fragrances has focused mainly on the physiological and biochemical content of the extraction process optimization of aroma volatiles [29], aroma component analysis [28], and in vitro activity evaluation of volatile oil extracts [30];”

This sentence is very confusing – what is the physiological content of a fragrance? What is the relevance of process optimisation? Isn’t ‘aroma component analysis’ not simply an analysis of VOCs? What are ‘volatile oil extracts’? There are essential oils, which are extracts and components of essential oils, which are (mostly) VOCs. Avoid confusing the reader by using multiple and unusual terms for the same thing (as mentioned earlier) and instead of excepted terminology resp.

Response: We are really sorry that we didn’t clearly represent the sentence. This sentence has been rewritten as “research on Nymphaea floral scents has focused mainly on the analysis of VOCs of different Nymphaea spp. and their relation with putative pollinators [28,29,30], the biosynthesis and emission of the major constituent of floral scent [31]”.

Comments 6: It would also help in this section to clarify the phylogeny of the species used and varieties/cultivars used, e.g. Nymphaea eldorado (by the way there is a persistent typo in the manuscript ‘eidorado’ instead of ‘eldorado’)

Response: We have added a description of Nymphaea ‘Eldorado’, and checked the plant names throughout the manuscript.

Results:

Comments 7: Overall, the section lacks focus – what is the main ‘take-home’ message? For example, the Venn diagram already shows components that are unique for each flowering stage – what is the purpose of the cluster analysis or PCA? What is the additional value/information that can be gained?

Response: According to the Reviewer’s comments, we have made modifications to make the results more focused. The Venn's diagram can directly shows the number of compounds shared and unshared/peculiaramong the different flowering stages of N. ‘Eldorado’ flowers, but it can't show which specific substances they are.

So in order to more clearly show the clustering of volatile substances in different periods and their differences, we added HCA and PCA, which can helps to determine the importance of parameters in the total variability through vector size, loads, and respective percent contributions. PCA can help to identify which volatiles contributed the most to the differences among the different flowering stages of N. ‘Eldorado’ flowers and for a better description of these scented plants.

Comments 8: Figures 2 and 9 do not provide any information beyond that presented in tables and other figures. I would leave those out, the more as the relative abundance scale makes absolute differences invisible.

Response: According to the suggestion of the reviewers, we have made appropriate adjustments. Figure 2 and 9 were deleted in the paper and put them in the supplementary materials (as Figure S1 and Figure S2) to make the article more concise.

Comments 9: Keep information of VOCs together – the SEM and TEM data are better presented after it is clear which part of the flower releases more VOCs

Response: The SEM and TEM data have been presented after the information of VOCs.

Comments 10: Line 158 – from the methods it appears that peak areas were normalised and should add up to 100 % for all four stages.

Response: When we analyzed the GC-MS data, a small amount of silicon-containing compounds were found in the volatile substances of each sample, which was analyzed as a systematic error caused by the internal environment of the machine, so the silicon-containing compounds were removed during the data analysis, so the total amount did not add up to 100%.

Comments 11: Line 159-160: It cannot be concluded that the content is larger in S3 when using normalised data. Equally between sample comparisons are valid but only show that relative concentrations are larger, which could be because e.g. fewer compounds are detected, other components are less abundant or the compound is actually more or less abundant. The total peak area provides an indication of overall increase or decrease of abundance of VOC.

Response: We have revised the sentence as suggested.

Comments 12: Line 174: unclear, what are “differences in the amount of substances with high volatile components”? Substances are all VOCs and the components mentioned in the following do not have particularly high vapour pressures.

Response: Sorry that we didn’t clearly represent the sentence. This sentence has been rewritten as “The concentration of the major volatile compounds varied in different flowering stages.”

Comments 13: Line 193 and following: this section is better placed earlier as it provides an overview of chemical classes and their relative abundance.

Response: We put this section after 2.1.1., because it was analyzed and summarized on the basis of the previous data.

Comments 14: Figure 1 – images of stages are in wrong order – S2 is last

Response: The images of stages were changed in right order, and we also replaced it with a more representative picture.

Discussion:

Comments 15: This section needs to be more concise and precise – avoid or be careful relating the work to areas that are not directly supported by your research, e.g. phylogeny, genetics, rapid changes in VOCs. Compare specific details, e.g. number of compounds, kind of compounds and avoid long paragraphs without references.

Response: It is really true as reviewer suggested and we revised the discussion part as suggested.

Comments 16: Lines: 315-320: true but the current research only looked at one species/cultivar.

Line 339: it is not clear why being a large tropical water lily is related to its VOC content and characteristics.

Response:We have deleted the irrelevant and general statements.

Comments 17: Line 339-362: this is essentially a repetition of results as there is no reference to the literature in the entire section.

Response: We have revised the discussion in this part, compared specific details, and added references.

Comments 18: Line 363: as earlier be consistent in terminology use ‘floral VOCs’ of VOCs – what constitutes the ‘aroma’?

Response: We have consistent in terminology use ‘floral VOCs’.

Material and Methods

Comments 19: The section is verbose in places – be more concise and precise

Line 485 and 495: delete ‘observations’

Response: We have revised the sentence as suggested,and deleted ‘observations’.

Comments 20: Line 498: what is meant by ‘The sample was lowered and vacuumed to sink,..”?

Response: The descriptions were removed.

Comments 21: Line 499: The sample was rinsed with phosphate buffer not the buffer

Response: We have revised the sentence as ”The sample was rinsed with phosphate buffer”.

Comments 22: Line 500: what acid was added? Was it added to make up to 0.1 % or was the concentration of added acid 0.1 %? What was the percentage of – volume or weight?

Response: The acid was 1% (w/w) osmic acid added in the same phosphate buffer.

Comments 23: Line 509: the fibre was conditioned not aged

Response: Fibres were conditioned at 250 °C for 30 min prior to use.

Comments 24: In terms of being more concise for example lines 508 to 517 could be written as “Floral VOC extraction: Floral VOCs were extracted from the headspace of 2.0 g of fresh flower petals placed into a 40 mL glass screw top vial (specifications) for 50 min at 45°C using a 65 µm PDMS/DVB SPME fibre (Supelco Inc, Bellefonte, USA) with manual handle. Samples were desorbed for 3 min at 2XX °C (specify!) in splitless mode (I assume?) in the injection port of a GC-MS for analysis (Trace 1300 ISO-LT GC1300, Thermofisher Scientific, Sunnyvale, CA, USA). Fibres were conditioned at 250 °C for 30 min prior to use.”

Response: We have revised the sentence as suggested.

Comments 25: How were VOCs collected from stamen and pistil? Were stamen and pistils separated from flowers (like the petals that are described)? Was the same weight used, same equipment etc?

Response: The stamen and pistils were separated from flowers like the petals with the same weight and the same equipment used.

Comments 26: Comments on the Quality of English Language

The English is overall of good quality but very verbose and inconsistent in technical terminology and jargon. It lacks conciseness, precision and consistence.

Response: We have edited the English language.

Special thanks to you for your good comments. We tried our best to improve the manuscript and made some changes in the manuscript. These changes will not influence the content and framework of the paper. We appreciate for your warm work earnestly, and hope that the correction will meet with approval.

Once again, thank you very much for your comments and suggestion.

If you have any queries, please don’t hesitate to contact us.

Thank you and best regards.

Sincerely yours

Reviewer 2 Report

Comments and Suggestions for Authors

Good paper, but many minor (systematic, partly) were found and should be corrected, as follows:

Use always italic for the name of investigated plant: Nympheae or N. Èidorado`

Add the voucher speciman for the botanical identification, including also the institution, where the speciman is stored

For (E)- and/or (Z)-isomers use allways the IUPAC-Rules: e.g. (Z)-5-dodecenyl-1 acetate; (Z6),(Z9)

As no chiral phase GC column was used, all (R)/(S)- and (+)/(-)- data must be eliminiated

Use linalool (not linasol) and 1,8-cineole (not cineole)

Table 1. Use always first big letter for the first adjectives at Aroma Description: e.g. No. 1: Wood, green, herb; No. 31: use Farnesene (not Famesene

Figure 5: Use 2-Phenyl-undecane (not 2-Phenyl-Undecane and 1-Methoxy-4-methyl-benzene (not 1-methoxy-4-methyl-Benzene

References: 17. Use short name of cited journal; 45. Name of the cited journal is unknown in international libraries

Author Response

Dear Reviewer:

Thank you for your comments concerning our manuscript entitled “Variation in the Floral Scent Chemistry of Nymphaea ‘Eldorado’, a Valuable Water Lily, With Different Flowering Stages and Flower Parts” (plants-2885832). Those comments are all valuable and very helpful for revising and improving our paper, as well as the important guiding significance to our researches. We have studied comments carefully and have made correction which we hope meet with approval. Revised portion are highlighted in the paper.

The main corrections in the paper and the responds to the comments are as following:

Response to Reviewer 2 Comments

Comments 1: Use always italic for the name of investigated plant: Nympheae or N. Èidorado`

Response: We have revised the name of investigated plant as suggested.

Comments 2: Add the voucher speciman for the botanical identification, including also the institution, where the speciman is stored

Response: We have added the voucher speciman for the botanical identification in 4.1. Plant Materials.

Comments 3: For (E)- and/or (Z)-isomers use allways the IUPAC-Rules: e.g. (Z)-5-dodecenyl-1 acetate; (Z6),(Z9)

As no chiral phase GC column was used, all (R)/(S)- and (+)/(-)- data must be eliminiated

Use linalool (not linasol) and 1,8-cineole (not cineole)

Response: We have corrected the chemical compounds in the manuscript.

Comments 4: Table 1. Use always first big letter for the first adjectives at Aroma Description: e.g. No. 1: Wood, green, herb; No. 31: use Farnesene (not Famesene

Response: We have revised the first big letter for the first adjectives at Aroma Description in Table 1.

Comments 5: Figure 5: Use 2-Phenyl-undecane (not 2-Phenyl-Undecane and 1-Methoxy-4-methyl-benzene (not 1-methoxy-4-methyl-Benzene

Response: We have revised the chemical compounds in Figure 5.

Comments 6: References: 17. Use short name of cited journal; 45. Name of the cited journal is unknown in international libraries

Response: We have used the short name of cited journal of the references 17 (Acta Hortic. Sin.) and revised the name of the cited journal of the references 45 (Sci. Hortic.).

Special thanks to you for your good comments. We tried our best to improve the manuscript and made some changes in the manuscript. These changes will not influence the content and framework of the paper. We appreciate for your warm work earnestly, and hope that the correction will meet with approval.

Once again, thank you very much for your comments and suggestion.

If you have any queries, please don’t hesitate to contact us.

Thank you and best regards.

Sincerely yours

Reviewer 3 Report

Comments and Suggestions for Authors

The work submitted for review is interesting. However, proofreading of the manuscript must be done before publication. The list of fixes is listed below:

- the abstract is too long and should be shortened;

- there are a lot of excess redundant spaces in the text and they should be removed;

- the acronym “HS-SPME-GC-MS” should be changed in the manuscript to “HS-SPME/GC-MS”;

- the nomenclature of chemical compounds requires correction to be consistent with IUPAC. Among other things, capital letters should be removed (e.g. 2-Thujene should be changed to 2-thujene).

- the description below table 1 requires correction because it does not refer to RI(exp.)

- the chapter: Extraction, Analysis and Detection of Volatile Floral Components requires correction. Please change “... and GC-MS analysis was performed by Trace 1300 ISO-LT GC1300 meteorological chromato- mass spectrometry (Thermo Fisher Scientific, Sunnyvale, CA, USA). to “... and gas chromatography - mass spectrometry analysis was performed by Trace 1300 ISQ LT (Thermo Scientific, Waltham, Massachusetts, USA).

- the text from lines 518 to 527 needs to be rewritten.

- in line 532 there are no references to the MS databases used.

Author Response

Dear Reviewer:

Thank you for your comments concerning our manuscript entitled “Variation in the Floral Scent Chemistry of Nymphaea ‘Eldorado’, a Valuable Water Lily, With Different Flowering Stages and Flower Parts” (plants-2885832). Those comments are all valuable and very helpful for revising and improving our paper, as well as the important guiding significance to our researches. We have studied comments carefully and have made correction which we hope meet with approval. Revised portion are highlighted in the paper.

The main corrections in the paper and the responds to the reviewers’ comments are as following:

Response to Reviewer 3 Comments

Comments 1: the abstract is too long and should be shortened;

Response: We have shortened the abstract as advised.

Comments 2: there are a lot of excess redundant spaces in the text and they should be removed;

Response: We have removed the excess redundant spaces in the whole text.

Comments 3: the acronym “HS-SPME-GC-MS” should be changed in the manuscript to “HS-SPME/GC-MS”;

Response: We have changed the “HS-SPME-GC-MS” to “HS-SPME/GC-MS” in the whole manuscript.

Comments 4: the nomenclature of chemical compounds requires correction to be consistent with IUPAC. Among other things, capital letters should be removed (e.g. 2-Thujene should be changed to 2-thujene).

Response: We have corrected the chemical compounds in the manuscript.

Comments 5: the description below table 1 requires correction because it does not refer to RI(exp.)

Response: We have corrected the description below table 1.

Comments 6: the chapter: Extraction, Analysis and Detection of Volatile Floral Components requires correction. Please change “... and GC-MS analysis was performed by Trace 1300 ISO-LT GC1300 meteorological chromato- mass spectrometry (Thermo Fisher Scientific, Sunnyvale, CA, USA).” to “... and gas chromatography - mass spectrometry analysis was performed by Trace 1300 ISQ LT (Thermo Scientific, Waltham, Massachusetts, USA).”

ResponseWe have revised the sentence as suggested.

Comments 7: the text from lines 518 to 527 needs to be rewritten.

Response: The text of this part has been rewritten as ”Transmission electron microscopy (TEM): First, petal samples from the full-flowering stage of N. 'Eldorado' were treated as rectangular pieces with a length of 5 mm and a width of 1 mm and then fixed in 4% glutaraldehyde prepared with 0.2 mol/L phosphoric acid buffer at pH=7.2 at 4 ℃ for 48 h. The sample was rinsed with phosphate buffer for 3 times, left for 10~20 min each time, then 3 h postfixation was performed in 1% (w/w) osmic acid in the same phosphate buffer, which was followed by dehydration with a graded series of acetone and embedding in Epon 812 epoxy resin. Finally, the embedded blocks were sliced with an LKB III ultramicrotome (LKB Instruments, Inc., AB Bromma, Stockholm, Sweden). The slices were stained with uranyl acetate and lead citrate, then photographed under a transmission electron microscope (Hitachi HU 12 A; Hitachi High-Technologies, Tokyo, Japan) [23].”.

Comments 8: in line 532 there are no references to the MS databases used.

Response: The references to the MS databases used were in the part of “4.4. Identification of Volatile Chemical Components.

Special thanks to you for your good comments. We tried our best to improve the manuscript and made some changes in the manuscript. These changes will not influence the content and framework of the paper. We appreciate for your warm work earnestly, and hope that the correction will meet with approval.

Once again, thank you very much for your comments and suggestion.

If you have any queries, please don’t hesitate to contact us.

Thank you and best regards.

Sincerely yours

Round 2

Reviewer 1 Report

Comments and Suggestions for Authors

Thank you very much for the thorough revision of the manuscript. It has improved a lot but I am afraid there is still room for improvement before the manuscript can be published.

Much of the corrections are about the use of English, e.g. direct articles, plural vs singular, prepositions, word use and phrasing.

There are, however, also still a few areas where the manuscript still lacks conciseness and precision; one remaining point is still the total relative content which is report as 96.01, 97.92 and 98.96 % - with HS-SPME-GC-MS only VOCs are recorded.

After removal of contaminants and artefacts, which are not constituents of the flower, the remaining total area is entirely composed of VOCs and should add up to 100 %

please find more specific details below:

Abstract:

Line 24: "A total of 60 volatile organic compounds were detected (VOCs) were ...

Line 27: "7 chemical classes"

Line 29: see comment above about %

Line 34: 'higher' instead of 'greater'

Line 41: delete 'What's more'

Line 53: 'main flowering stage' location can only be suggested and use floral VOCs instead of substances as only VOCs were investigated.

Introduction

Line 60: delete ‘The’ and ‘many’: “Floral scents are composed of volatile organic compounds…”

Line 64: delete ‘The’: “Floral VOCs have….”

Line 69: delete ‘different’: “….VOC release differ among species and varieties…”

Line 77: what is ‘dilute basil’?

Line 80: it would be sufficient just to state: “….few studies on VOCs from water lilies, a well-know aquatic flowering plant.”

Line 93: it is not clear what is meant by “The components with higher volatile contents..”

Line 118: use plural “water lilies”

Line 119: delete ‘different’ (to differ means to be different, hence no need for an extra ‘different’)

Line 127: delete ‘the’; change ‘in’ to ‘at’ and add ‘in’: “…done on VOCs at different flowering stages and in floral organs of water lilies.”

Line 131: maybe better ‘aromatic compounds’ instead of ‘benzene ring compounds’

Line 134: change to: “….with alkanes being the most common.”

Line 137: alkanes appear twice

Line 145-146: change to “Chinese horticulturists cross-bred water lilies for many years…”

Results: 

Line 194: Leave out the 1st sentence of the paragraph, just refer to figure 1

Line 208: better say ‘… floral scent consisted mainly of…”

Line 212-213: correct to “Content of benzaldehyde…significantly higher at the…”

Line 215: as in line 208 better say “…the aroma consisted mainly of…”

Figure 2 – Legend: it is numbers not amount of VOCs at different stages

Line 228 and following: they are classes of compounds

Line 229: “Of the 7 classes, alkenes (28) accounted…”

Line 230: just ‘alkanes’ not ‘alkane compounds’

Line 231: just “(8)”

Line 237, 241: use ‘higher’ rather than ‘greater’

Line 243: “VOCs from all classes were present in some proportion” instead of ‘all kinds of volatile substances”

Figure 4: what is the unit of the scale? Provide a little detail how it relates to relative concentrations reported in Table 1

Line 262 – 285: there is no need to explain what was done and how in this section  - present what was found. In the event positive or negative PC values are less relevant than the direction towards individual stages.

Figure 5: describe what the ellipses represent.

Line 316: delete 1st sentence of paragraph.

Line 318: the structure changes through stages of flower development – moreover, the changes are causative for the release of VOCs and not the other way round.

Line 318-320: Write simple, e.g. ‘No papillae were evident in the bud stage of petals of N Eldorado (Fig 8A-S1).

Line 324-325: delete “the petals of N. Eldorada gradually unfolded”

Line 326: ‘plastids’ not ‘plasts’

Line 327: what are “matrix particles of osmophilic acid”?

Line 329-330: delete “the petals of the N. ‘E_idoradoldorado’ _flower fully unfolded, and the”

Line 331: correct to “volume of plastids”

Line 335-336: delete “the petals of N. ‘E_idoradoldorado’ _fell off when touched by hands, and the flower organs gradually shrunk. At this time, the”

Line 337: again ‘plastids’ not ‘plasts’

Line 340: better use ‘”… results suggest a relationship between cellular structure…”

Discussion:

Generally, speak about VOCs when that is wat is meant by ‘volatile substances’ or ‘floral substances’ be consistent with terminology.

Line 392-393: what is meant by ‘floral substances present at this stage are the most complete..’?

Line 393: the statement ‘all kinds of substances decline sharply…’ is too imprecise. This could mean anything from VOCs, primary metabolites to proteins peptides and structural components – specify or exemplify here  (and in following) what’s reduced and by how much (e.g. 90 % reduction).

Line 397-402: relevance not clear.

Line 446-458: the section is relevant but in the absence of any observation about pollinators also rather speculative, mention but reduce size.

Line 467-482: this is essentially a repetition of the results – compare and contrast directly with e.g. ref 48 ad 49.

Line 514: pistillate and staminate phase come up for the first time – relate those to the flowering stages.

Line 524-537: the relevance of the section is not clear – transcriptomic analysis did not target VOC forming pathways and any connection is very speculative.

Figure S1 please correct name of species to N. Eldorado

Comments on the Quality of English Language

Please refer to my comments above, the English language issues are of a nature that escapes software tools.

Author Response

Thank you for your comments concerning our manuscript entitled “Variation in the Floral Scent Chemistry of Nymphaea ‘Eldorado’, a Valuable Water Lily, With Different Flowering Stages and Flower Parts” (plants-2885832). Those comments are all valuable and very helpful for revising and improving our paper, as well as the important guiding significance to our researches. We have studied comments carefully and have made correction which we hope meet with approval. Revised portion are highlighted in the paper.

The main corrections in the paper and the responds to the reviewer’s comments are as following:

Response to Reviewer 1 Comments

Thank you very much for the thorough revision of the manuscript. It has improved a lot but I am afraid there is still room for improvement before the manuscript can be published.

Much of the corrections are about the use of English, e.g. direct articles, plural vs singular, prepositions, word use and phrasing.

Response: We are really sorry that the revised manuscript has not meet the requirements. We tried to improve the manuscript and made major revisions in the manuscript which we hope meet with approval.

There are, however, also still a few areas where the manuscript still lacks conciseness and precision; one remaining point is still the total relative content which is report as 96.01, 97.92 and 98.96 % - with HS-SPME-GC-MS only VOCs are recorded.

After removal of contaminants and artefacts, which are not constituents of the flower, the remaining total area is entirely composed of VOCs and should add up to 100 %

Response: It is true as the reviewer said that the total relative content of VOCs should be 100%. In fact, since some VOCs with very low relative content (<0.05%) and the not identified compound were not counted in the statistics of volatile substances of the flowers, we counted VOCs with relative content higher than 0.05%, so the total relative content was not 100%. We add these explanation below Table 1. Some other studies of plant VOCs have used the same method, such as (Merajuddin Khan, 2023; Milica Acimovic, 2021; Yun-Su Baek, 2019; etc.). We hope our explanation can resolve these doubts, if it still not convincing, we will continue to revise it .

please find more specific details below:

Abstract:

Line 24: "A total of 60 volatile organic compounds were detected (VOCs) were ...

Line 27: "7 chemical classes"

Line 29: see comment above about %

Line 34: 'higher' instead of 'greater'

Line 41: delete 'What's more'

Line 53: 'main flowering stage' location can only be suggested and use floral VOCs instead of substances as only VOCs were investigated.

Response: We have revised all the sentences in the Abstract as the reviewer suggested. 

Introduction

Line 60: delete ‘The’ and ‘many’: “Floral scents are composed of volatile organic compounds…”

Line 64: delete ‘The’: “Floral VOCs have….”

Line 69: delete ‘different’: “….VOC release differ among species and varieties…”

Line 77: what is ‘dilute basil’?

Response: Sorry, it is ocimene.

Line 80: it would be sufficient just to state: “….few studies on VOCs from water lilies, a well-know aquatic flowering plant.”

Line 93: it is not clear what is meant by “The components with higher volatile contents..”

Response: The sentence was rewritten as “Research has shown that the total volatile content in Lilium spp. flowers at the full flowering stage was higher than that at the initial flowering stage, with a high content of linalsol, (Z)-β-basil, 1,8-cineole, (+) limonene, laurene, methyl benzoate, etc. [17,18]”.

Line 118: use plural “water lilies”

Line 119: delete ‘different’ (to differ means to be different, hence no need for an extra ‘different’)

Line 127: delete ‘the’; change ‘in’ to ‘at’ and add ‘in’: “…done on VOCs at different flowering stages and in floral organs of water lilies.”

Line 131: maybe better ‘aromatic compounds’ instead of ‘benzene ring compounds’

Line 134: change to: “….with alkanes being the most common.”

Line 137: alkanes appear twice

Line 145-146: change to “Chinese horticulturists cross-bred water lilies for many years…”

Response: We have revised all the sentences in the Introduction part as the reviewer suggested. 

Results: 

Line 194: Leave out the 1st sentence of the paragraph, just refer to figure 1

Line 208: better say ‘… floral scent consisted mainly of…”

Line 212-213: correct to “Content of benzaldehyde…significantly higher at the…”

Line 215: as in line 208 better say “…the aroma consisted mainly of…”

Figure 2 – Legend: it is numbers not amount of VOCs at different stages

Line 228 and following: they are classes of compounds

Line 229: “Of the 7 classes, alkenes (28) accounted…”

Line 230: just ‘alkanes’ not ‘alkane compounds’

Line 231: just “(8)”

Line 237, 241: use ‘higher’ rather than ‘greater’

Line 243: “VOCs from all classes were present in some proportion” instead of ‘all kinds of volatile substances”

Figure 4: what is the unit of the scale? Provide a little detail how it relates to relative concentrations reported in Table 1

Response: We add “Center scaled value” under the legend in Figure 4, the scale has no units, which was calculated by the “scale” function. The heat map was normalized using the row scale method.

Line 262 – 285: there is no need to explain what was done and how in this section  - present what was found. In the event positive or negative PC values are less relevant than the direction towards individual stages.

Response: We have revised the sentences in this section, and deleted the redundant sentences. 

Figure 5: describe what the ellipses represent.

Response: We have added a description of ellipses in Figure 5. The ellipses represented samples of the same stage and the VOCs that were highly correlated with that stage.

Line 316: delete 1st sentence of paragraph.

Line 318: the structure changes through stages of flower development – moreover, the changes are causative for the release of VOCs and not the other way round.

Line 318-320: Write simple, e.g. ‘No papillae were evident in the bud stage of petals of N Eldorado (Fig 8A-S1).

Line 324-325: delete “the petals of N. Eldorada gradually unfolded”

Line 326: ‘plastids’ not ‘plasts’

Line 327: what are “matrix particles of osmophilic acid”?

Response: We have changed “matrix particles of osmophilic acid” to “osmophilic matrix granules”.

Line 329-330: delete “the petals of the N. ‘E_idoradoldorado’ _flower fully unfolded, and the”

Line 331: correct to “volume of plastids”

Line 335-336: delete “the petals of N. ‘E_idoradoldorado’ _fell off when touched by hands, and the flower organs gradually shrunk. At this time, the”

Line 337: again ‘plastids’ not ‘plasts’

Line 340: better use ‘”… results suggest a relationship between cellular structure…”

Response: We have revised all the sentences in the Results part as the reviewer suggested. 

Discussion:

Generally, speak about VOCs when that is wat is meant by ‘volatile substances’ or ‘floral substances’ be consistent with terminology.

Response: We have changed ‘volatile substances’ or ‘floral substances’ to ‘VOCs ‘.

Line 392-393: what is meant by ‘floral substances present at this stage are the most complete..’?

Response: We have deleted this sentence.

Line 393: the statement ‘all kinds of substances decline sharply…’ is too imprecise. This could mean anything from VOCs, primary metabolites to proteins peptides and structural components – specify or exemplify here  (and in following) what’s reduced and by how much (e.g. 90 % reduction).

Response: We have specified and exemplified what’s reduced and by how much in these statements. The sentence has been rewritten as “the release of VOCs and its diversity are lower at the bud stage and increase sharply at the blooming stage, then VOCs decline sharply after entering the final flowering stage. For example, more hydrocarbon, esters, and alcohols compounds were detected at the full-flowering stage than other flower stages of C. odorata, the total number of VOCs at wilted-flowering stage decreased nearly half of that at full-flowering stage [40]. ”

Line 397-402: relevance not clear.

Response: We have deleted the irrelevant statements.

Line 446-458: the section is relevant but in the absence of any observation about pollinators also rather speculative, mention but reduce size.

Response: We have reduced the size of the observation about pollinators.

Line 467-482: this is essentially a repetition of the results – compare and contrast directly with e.g. ref 48 ad 49.

Response: We have deleted the repetition of the results, compared and contrasted directly with the references in 3.2. Differences in Petal Structure at Different Flowering Stages and Their Relationships with Floral Release.

Line 514: pistillate and staminate phase come up for the first time – relate those to the flowering stages.

Response: We have relate the pistillate and staminate phase to the flowering stages in the second paragraph of 3.1. Differences in Floral Components of N. 'Eldorado' at Different Flowering Stages.

Line 524-537: the relevance of the section is not clear – transcriptomic analysis did not target VOC forming pathways and any connection is very speculative.

Response: We have revised the sentences in this section to make the content more relevant.

Figure S1 please correct name of species to N. Eldorado

Response: We have corrected the name of species to N. Eldorado’ in Figure S1.

Special thanks to you for your good comments. We tried our best to improve the manuscript and made some changes in the manuscript. These changes will not influence the content and framework of the paper. We appreciate for your warm work earnestly, and hope that the correction will meet with approval.

Once again, thank you very much for your comments and suggestion.

If you have any queries, please don’t hesitate to contact us by this e-mail address: zhuzunling@njfu.edu.cn.

Thank you and best regards.

Sincerely yours

Reviewer 2 Report

Comments and Suggestions for Authors

Some mistakes were detected, as follows:

Use always italic for the investigated plant: Nymphaea and N. ´Eidorado´

As you did not use a chiral phase GC-column, no stereochemical informations can be given and therefore eliminated wherever stated: (+)/(-)- and/or (R)/(S)-isomers

For (Z)- and (E)-compounds use always: e.g. (Z)-5-dodecen-1-yl acetate

Some names of identified compounds in the figures, tables and/or text are given wrong: linalool (not linasol), 1,8-cineole (not cineole), 2-Phenyl-undecane (not 2-Phenyl-Undecane), 1-Methoxy-4-methyl-benzene (not 1-methoxy-4-methyl-Benzene), 1,3-Cyclohexadiene,1-(1,5-dimethyl-hexen-1-yl)-4-methyl is an unknown name of compound

In Table 1, Aroma Description:

use woody (not wood), herbal (not herb), spicy (not spice)

use always big letter for the first attribute: e.g. 1 Woody, green, herb

At the part 4.1. Plant Materials: add voucher specimen of the botanical identification of the investigated plant, including the institution, where the specimen is stored

In the section References:

17. use short name of cited jounal

45. name of cited jounal is unknown ( make a library-such for the correct name)

Author Response

Dear Reviewer:

Thank you for your comments concerning our manuscript entitled “Variation in the Floral Scent Chemistry of Nymphaea ‘Eldorado’, a Valuable Water Lily, With Different Flowering Stages and Flower Parts” (plants-2885832). Those comments are all valuable and very helpful for revising and improving our paper, as well as the important guiding significance to our researches. We have studied comments carefully and have made correction which we hope meet with approval. Revised portion are highlighted in the paper.

The main corrections in the paper and the responds to the reviewer’s comments are as following:

Response to Reviewer 2 Comments

Some mistakes were detected, as follows:

Point 1: Use always italic for the investigated plant: Nymphaea and N. ´Eidorado´

Response: We have revised the name of investigated plant as suggested, all Nymphaea and N. ´Eldorado´ were italic.

Point 2: As you did not use a chiral phase GC-column, no stereochemical informations can be given and therefore eliminated wherever stated: (+)/(-)- and/or (R)/(S)-isomers, For (Z)- and (E)-compounds use always: e.g. (Z)-5-dodecen-1-yl acetate

Response: We have eliminated the (+)/(-)- and/or (R)/(S)-isomers, and revised the (Z)- and (E)-compounds in the whole manuscript: e.g. (Z6),(Z9)-Pentadecadien-1-ol; (Z)-5-Dodecenyl-1 acetate.

Point 3: Some names of identified compounds in the figures, tables and/or text are given wrong: linalool (not linasol), 1,8-cineole (not cineole), 2-Phenyl-undecane (not 2-Phenyl-Undecane), 1-Methoxy-4-methyl-benzene (not 1-methoxy-4-methyl-Benzene),1,3-Cyclohexadiene,1-(1,5-dimethyl-4-hexen-1-yl)-4-methyl is an unknown name of compound

Response: We have corrected the chemical compounds in the manuscript, changed “linasol” to “linalool”, “cineole” to “1,8-cineole”, “2-Phenyl-Undecane” to “2-Phenyl-undecane”, “1-methoxy-4-methyl-Benzene” to “1-Methoxy-4-methyl-benzene”. 

The name “1,3-Cyclohexadiene,1-(1,5-dimethyl-4-hexen-1-yl)-4-methyl” can write as “1-(1,5-Dimethyl-4-hexen-1-yl)-4-methyl-1,3-cyclohexadiene”, its CAS No.: 451-55-8.

Point 4: In Table 1, Aroma Description:use woody (not wood), herbal (not herb), spicy (not spice), use always big letter for the first attribute: e.g. 1 Woody, green, herb

Response: We have revised the Aroma Description (use woody, herbal, and spicy), and used the big letter for the first adjectives in Table 1: e.g.

Point 5: At the part 4.1. Plant Materials: add voucher specimen of the botanical identification of the investigated plant, including the institution, where the specimen is stored

Response: We have added the voucher speciman for the botanical identification.This sentence has been rewritten as “A voucher specimen (No.20181025) was deposited in the Institute of Nanjing Forestry University. ”in 4.1. Plant Materials.

Point 6: In the section References:

  1. use short name of cited jounal
  2. name of cited jounal is unknown ( make a library-such for the correct name)

Response: We have revised the names of cited journal of the references.

  • 17(Huixiu, Z.; Pingsheng, L.; Zenghui, H.U.; Jing, Z.; Wenhe, W.; Fang, A.X. The floral scent emitted from 'Siberia'at different flowering stages and diurnal v Acta Hortic. Sin2013, 40, 693-702. https://doi.org/10.16420/j.issn.0513-353x.2013.04.012 ); 

(2) 45 (Yonglu, M.; Na, L.; Ji, T.; Junping, G.; Changqing, Z. Identification and validation of reference genes for gene expression studies in postharvest rose flower (Rosa hybrida). Sci. Hortic2013158, 16-21. https://doi.org/10.1016/j.scienta.2013.04.019).

Special thanks to you for your good comments. We tried our best to improve the manuscript and made some changes in the manuscript. These changes will not influence the content and framework of the paper. We appreciate for your warm work earnestly, and hope that the correction will meet with approval.

Once again, thank you very much for your comments and suggestion. If you have any queries, please don’t hesitate to contact us by this e-mail address: zhuzunling@njfu.edu.cn.

Thank you and best regards.

Sincerely yours

Reviewer 3 Report

Comments and Suggestions for Authors

Unfortunately, the text submitted for review has not been corrected in accordance with previous suggestions.

The use of capital letters in compound names has not been standardized. This applies to the names of compounds that are still incorrect in both Table 1 and Figure 4 (e.g. is (E)-α-Famesene, should be (E)-α-Farnesene).

Chapter 4.3. Extraction, Analysis and Detection of Volatile Floral Components is not suitable for publication in its current form. The GC-MS name is still incorrect (is Trace 1300 ISO-LT and should be Trace 1300 ISQ LT). GC-MS is not produced by Thermofisher Scientific but by Thermo Scientific. The text of this chapter is also not written in a scientific form and requires correction. The chapter also does not provide information whether the SPME fiber was conditioned before analysis. If it was conditioned, under what conditions. There are still no citations to the MS databases used (NIST, PubChem, and Pherobase).

Author Response

Dear Reviewer:

Thank you for your comments concerning our manuscript entitled “Variation in the Floral Scent Chemistry of Nymphaea ‘Eldorado’, a Valuable Water Lily, With Different Flowering Stages and Flower Parts” (plants-2885832). Those comments are all valuable and very helpful for revising and improving our paper, as well as the important guiding significance to our researches. We have studied comments carefully and have made correction which we hope meet with approval. Revised portion are highlighted in the paper.

The main corrections in the paper and the responds to the reviewer’s comments are as following:

Response to Reviewer 3 Comments

Unfortunately, the text submitted for review has not been corrected in accordance with previous suggestions.

Response: We are really sorry that the revised manuscript has not meet the requirements. We tried to improve the manuscript and made major revisions in the manuscript.

Point 1: The use of capital letters in compound names has not been standardized. This applies to the names of compounds that are still incorrect in both Table 1 and Figure 4 (e.g. is (E)-α-Famesene, should be (E)-α-Farnesene).

Response:

(1)We have standardized the capital letters in compound names according to the comments from reviewers, and also refer to the format of the compound names in the table from the previous papers published in the journal of Plants (e.g. Merajuddin Khan, 2023; Leonardo Llorens, 2023.).

(2)We have changed the “(E)-α-Famesene” to “(E)-α-Farnesene” in both Table 1 and Figure 4 .

Point 2: Chapter 4.3. Extraction, Analysis and Detection of Volatile Floral Components is not suitable for publication in its current form. The GC-MS name is still incorrect (is Trace 1300 ISO-LT and should be Trace 1300 ISQ LT). GC-MS is not produced by Thermofisher Scientific but by Thermo Scientific. The text of this chapter is also not written in a scientific form and requires correction.

The chapter also does not provide information whether the SPME fiber was conditioned before analysis. If it was conditioned, under what conditions.

There are still no citations to the MS databases used (NIST, PubChem, and Pherobase).

Response: 

(1) We have revised the sentence as suggested, the sentence has been rewritten as “ After extraction, the fiber was desorbed for 3 min at 250 °C in splitless mode in the injection port, and gas chromatography-mass spectrometry was performed by Trace 1300 ISQ LT (Thermo Scientific, Waltham, Massachusetts, USA). ”. 

The text of this chapter was written as:

 “4.3. Analysis of Volatile Components

Floral VOCs were extracted from the headspace of 2.0 g of fresh flower samples (petals/stamen/pistil) placed into a 40 mL glass screw top vial (specifications) for 50 min at 45°C using a 65 µm PDMS/DVB SPME fibre (Supelco Inc., Bellefonte, USA) with manual handle. After extraction, the fiber was desorbed for 3 min at 250 °C in splitless mode in the injection port, and gas chromatography-mass spectrometry was performed by Trace 1300 ISQ LT (Thermo Scientific, Waltham, Massachusetts, USA). The SPME fibre was conditioned at 250 °C for 30 min before first use.

A DB-5 MS capillary column (5% phenylmethylsiloxane, 30 m×0.25 mm×0.25 μm; Thermo Fisher Scientific, Sunnyvale, CA, USA) was used to conduct the analyses. Helium was used as a carrier gas with a flow rate of 1.0 mL/min without diverting. The inlet temperature was 250°C. The oven temperature was programmed at 40°C for 2 min, then raised at 4°C/min to 110°C and held for 2 min. Then the temperature was raised at 3°C/min to 150°C and held for 2 min, then increased at 5°C/min to 200°C and held for 4 min. The column flow rate was 1.0 mL/min in splitless mode. The mass selective detector was used in EI mode at an ionization voltage of 70 eV, and the scanning range was m/z 33–450 in the full-scan mode. The ion source and quadrupole temperatures were 250 °C.

Retention indices (RIs) were calculated by using the retention times of C7–C30 n-alkanes (BNCC, China) according to previously reported methods [33, 58]. The volatile components were identified by comparison of their RI and mass spectra with the NIST (National Institute of Standards and Technology, https://webbook.nist.gov/chemistry/), PubChem (https://pubchem.ncbi.nlm.nih.gov/) and Pherobase (www.pherobase.com) databases [59,60], and the relevant literature [29,30,42,61,62]. The relative content of each component in the sample was calculated by the peak area normalization method [63]. The spectrum of each compound was analyzed by Xcalibur and NIST 2014 (NIST Database, ChemSW, Inc., Fairfield, CA, USA), and compounds were identified according to their retention times (RTs) and NIST database.”.

(2)We have added the information of the conditions of the SPME fiber, the sentence has been rewritten as“The SPME fibre was conditioned at 250 °C for 30 min before first use.”

(3) We have added the citations to the MS databases used, the sentence has been rewritten as “The volatile components were identified by comparison of their RI and mass spectra with the NIST (National Institute of Standards and Technology, https://webbook.nist.gov/chemistry/), PubChem (https://pubchem.ncbi.nlm.nih.gov/) and Pherobase ( www.pherobase.com) databases [59,60], and the relevant literature [29,30,42,61,62]. ”.

Special thanks to you for your good comments. We tried our best to improve the manuscript and made some changes in the manuscript. These changes will not influence the content and framework of the paper. We appreciate for your warm work earnestly, and hope that the correction will meet with approval.

Once again, thank you very much for your comments and suggestion.

If you have any queries, please don’t hesitate to contact us by this e-mail address: zhuzunling@njfu.edu.cn.

Thank you and best regards.

Sincerely yours

Round 3

Reviewer 2 Report

Comments and Suggestions for Authors

Some minor corrections necessary, as follows:

Use always linalool (not linasol),

Use always (6Z,9Z-pentdecadien-1ol and (E6,,E9)-heptadecadiene

Author Response

Dear Reviewer:

Thank you for your comments concerning our manuscript entitled “Variation in the Floral Scent Chemistry of Nymphaea ‘Eldorado’, a Valuable Water Lily, With Different Flowering Stages and Flower Parts” (plants-2885832). Those comments are all valuable and very helpful for revising and improving our paper, as well as the important guiding significance to our researches. We have studied comments carefully and have made correction which we hope meet with approval. Revised portion are highlighted in the paper.

The main corrections in the paper and the responds to the reviewer’s comments are as following:

Response to Reviewer 2 Comments

Some minor corrections necessary, as follows:

Use always linalool (not linasol),

Response: We have changed “linasol” to “linalool” in the Introduction part, and also checked the full text to make sure that there were no other similar errors.

Point 2: Use always (6Z,9Z-pentdecadien-1ol and (E6,E9)-heptadecadiene

Response: We have used (6Z,9Z)-pentdecadien-1-ol and (E6,E9)-heptadecadiene in the whole paper.

Once again, thank you very much for your comments and suggestion. If you have any queries, please don’t hesitate to contact us by this e-mail address: zhuzunling@njfu.edu.cn.

Thank you and best regards.

Sincerely yours

Reviewer 3 Report

Comments and Suggestions for Authors

Most of the suggestions included in the review were taken into account by the authors. In its current form, the text requires only minor text corrections. The fixes are below:

Section 4.3. Analysis of Volatile Components

Floral VOCs were extracted from the headspace of 2.0 g of fresh flower samples (petals/stamen/pistil) placed into a 40 mL glass screw top vial (specifications) for 50 min at 45 ° C using a 65 μm PDMS/DVB SPME fibre (Supelco Inc., Bellefonte, USA) with manual handle. After extraction, the fiber was desorbed for 3 min at 250 °C in splitless mode in the injection port, and gas chromatography-mass spectrometry was performed by Trace 1300 ISQ LT (Thermo Scientific, Waltham, Massachusetts, USA). The SPME fibre was conditioned at 250 °C for 30 min before first use. A DB-5 MS capillary column (5% phenylmethylsiloxane, 30 m×0.25 mm×0.25 μm; Thermo Fisher Scientific, Sunnyvale, CA, USA) was used to conduct the analyses. Helium was used as a carrier gas with a flow rate of 1.0 mL/min without diverting. The inlet temperature was 250°C. The oven temperature was programmed at 40°C for 2 min, then raised at 4°C/min to 110°C and held for 2 min. Then the temperature was raised at 3°C/min to 150°C and held for 2 min, then increased at 5°C/min to 200°C and held for 4 min. The column flow rate was 1.0 mL/min in splitless mode. The mass selective detector was used in EI mode at an ionization voltage of 70 eV, and the scanning range was m/z 33–450 in the full-scan mode. The ion source and quadrupole temperatures were 250 °C.

References to MS databases placed in lines 452 and 453 should be moved to the References section.

Author Response

Dear Reviewer:

Thank you for your comments concerning our manuscript entitled “Variation in the Floral Scent Chemistry of Nymphaea ‘Eldorado’, a Valuable Water Lily, With Different Flowering Stages and Flower Parts” (plants-2885832). Those comments are all valuable and very helpful for revising and improving our paper, as well as the important guiding significance to our researches. We have studied comments carefully and have made correction which we hope meet with approval. Revised portion are highlighted in the paper.

The main corrections in the paper and the responds to the reviewer’s comments are as following:

Response to Reviewer 3 Comments

Most of the suggestions included in the review were taken into account by the authors. In its current form, the text requires only minor text corrections. The fixes are below:

Section 4.3. Analysis of Volatile Components

Floral VOCs were extracted from the headspace of 2.0 g of fresh flower samples (petals/stamen/pistil) placed into a 40 mL glass screw top vial (specifications) for 50 min at 45 ° C using a 65 μm PDMS/DVB SPME fibre (Supelco Inc., Bellefonte, USA) with manual handle. After extraction, the fiber was desorbed for 3 min at 250 °C in splitless mode in the injection port, and gas chromatography-mass spectrometry was performed by Trace 1300 ISQ LT (Thermo Scientific, Waltham, Massachusetts, USA). The SPME fibre was conditioned at 250 °C for 30 min before first use. A DB-5 MS capillary column (5% phenylmethylsiloxane, 30 m×0.25 mm×0.25 μm; Thermo Fisher Scientific, Sunnyvale, CA, USA) was used to conduct the analyses. Helium was used as a carrier gas with a flow rate of 1.0 mL/min without diverting. The inlet temperature was 250°C. The oven temperature was programmed at 40°C for 2 min, then raised at 4°C/min to 110°C and held for 2 min. Then the temperature was raised at 3°C/min to 150°C and held for 2 min, then increased at 5°C/min to 200°C and held for 4 min. The column flow rate was 1.0 mL/min in splitless mode. The mass selective detector was used in EI mode at an ionization voltage of 70 eV, and the scanning range was m/z 33–450 in the full-scan mode. The ion source and quadrupole temperatures were 250 °C.

Response: We have revised this part as the reviewer suggested, the sentences  marked in red were changed to “Helium was used as a carrier gas with a flow rate of 1.0 mL/min. The injector temperature was 250°C in splitless mode. ”; ”The flow rate was 1.0 mL/min”.

Point 2: References to MS databases placed in lines 452 and 453 should be moved to the References section.

Response: We have moved the References to MS databases placed in lines 452 and 453 to the References section.

They were:

  1. NIST Chemistry WebBook. Available online: https://webbook.nist.gov/chemistry/ (accessed on 22 September 2023).
  2. PubChem. Available online: https://pubchem.ncbi.nlm.nih.gov/ (accessed on 22 September 2023).
  3. The Pherobase: Database of pheromones and semiochemicals. Available online: https://www.pherobase.com (accessed on 25 September 2023).

Once again, thank you very much for your comments and suggestion. If you have any queries, please don’t hesitate to contact us by this e-mail address: zhuzunling@njfu.edu.cn.

Thank you and best regards.

Sincerely yours
